



# Modelling the Spectral Shape of Continuous-Wave Lidar Measurements in a Turbulent Wind Tunnel

Marijn Floris van Dooren[1], Anantha Padmanabhan Kidambi Sekar[1], Lars Neuhaus[1], Torben Mikkelsen[2], Michael Hölling[1], and Martin Kühn[1]

[1]ForWind – Centre for Wind Energy Research, Institute of Physics, University of Oldenburg, Küpkersweg 70, 26129 Oldenburg, Germany
[2]Technical University of Denmark, Department of Wind Energy, Frederiksborgvej 399, 4000 Roskilde, Denmark

**Correspondence:** Marijn Floris van Dooren (marijn.vandooren@uol.de)

**Abstract.** This paper describes the development of a model for the turbulence spectrum measured by a short-range, continuous-wave lidar. The lidar performance was assessed by measurements conducted with two WindScanners in an open jet wind tunnel equipped with an active grid, for a range of different turbulent wind conditions. A one-dimensional hot wire anemometer was used as a reference for characterising the lidar turbulence measurement. In addition to addressing the statistics, the correlation
between the time series and the mean error on the wind speed, the lidar measurement turbulence spectrum is compared with a theoretical spectrum using Taylor's frozen turbulence hypothesis. A theoretical model for the probe length averaging effect is applied in the frequency domain, using a Lorentzian filter in combination with generated white noise, and evaluated by qualitatively matching the lidar measurement spectrum. High goodness of fit coefficients and low mean absolute errors between hot wire and WindScanner were observed for the measured time series. The correlation showed an inverse relationship with
the prevalent turbulence intensity in the flow for cases with a comparable power spectrum shape. Larger flow structures can be captured more accurately by the lidar, whereas small-scale turbulent flow structures are partly filtered out as a result of the lidars' probe volume averaging. It is demonstrated that an accurate way to define the frequency at which the lidar power spectrum starts to deviate from the hot wire reference spectrum is the point at which the coherence drops below 0.5. This coherence-based cut-off frequency increases linearly with the mean wind speed and is generally an order of magnitude lower
than the probe length cut-off frequency, estimated according to a simple model based on Taylor's frozen turbulence hypothesis. A convincing match between the modelled and the actual WindScanner power spectrum was found for various different cases, which confirmed that the deviation of the lidar measurement power spectrum in the higher frequency range can be analytically explained and modelled as a combination of a Lorentzian probe length averaging effect and white noise in the lidar measurement.

## 1   Introduction

Wind tunnels are frequently used to reproduce wind conditions more realistically, e.g. in order to perform meaningful model wind turbine tests. Existing wind tunnels can simulate the atmospheric boundary layer through passive flow manipulation by means of roughness elements and spires, one example being the facility at the Polytechnic University of Milan (Bottasso et al.,





2014). Other wind tunnels, such as the wind tunnel of the University of Oldenburg, are equipped with an active grid which can
generate user-defined wind conditions, e.g. wind shear and turbulence (Kröger et al., 2018; Neuhaus et al., 2020, 2021).

These complex wind conditions require equally sophisticated ways to measure the flow in the wind tunnel. Hot wire anemometers (Bradshaw, 1971; Comte-Bellot, 1976) have a high temporal resolution and can measure one to three wind speed components at a single point in space. Particle image velocimetry (PIV) can be used to measure a highly resolved two-dimensional wind flow over a small area of interest (Adrian and Westerweel, 2007). Laser Doppler anemometers (LDA) use the
Doppler effect in order to remotely measure one, two or three wind speed components by focusing a monochromatic, coherent laser at a single point in space (Durst et al., 1976).

The continuous-wave (cw) lidar (Slinger and Harris, 2012; Pitter et al., 2013) is also a type of laser-based Doppler anemometer. Lidars measure the line-of-sight component of the wind speed and have a slightly lower temporal and spatial resolution than the two aforementioned sensors, but make up for it with a non-invasive and flexible scanning measurement technique. The
execution of lidar measurements inside a wind tunnel (Pedersen et al., 2012; van Dooren et al., 2017; Sjöholm et al., 2017; Hulsman et al., 2020) is a relatively new and challenging concept, which we believe to be a future benchmark for the measurement of the flow through and at cross-sections inside a wind tunnel, either with or without model wind turbines interacting with the flow.

Previous research on the effect of the lidar's effective probe length averaging and range gating on the measurement spectrum
focuses mainly on the spectral transfer function. Angelou et al. (2012) propose two different ways to model the spectral transfer function between lidar and sonic anemometer time series. Held and Mann (2018) investigate the effect of the line-of-sight velocity determination method on the shape of the spectral transfer function. Puccioni and Iungo (2021) propose a spectral correction for the range-gate averaging effect of pulsed lidars, based on a simple low-pass filter fitted to the lidar's frequency spectrum. In this paper we implement a similar approach for a continuous-wave lidar, however, we use an analytical
approach to model the lidar's measurement spectrum, that includes a minimal amount of a-priori information from the lidar measurement itself.

Consequently, the main objective of this paper is to address the performance and the limitations of using short-range lidar for turbulence measurements, by modelling, comparing and evaluating the measurement principle in the frequency domain. This is addressed by analysing the following three aspects;

1. the correlation between the lidar and hot wire time series, quantified with the linear regression goodness of fit coefficient,

2. the error on the mean wind speed, quantified as the difference between the cw-lidar and hot wire mean wind speed,

3. the comparison between the cw-lidar and hot wire measurement in the frequency domain, quantified by two different definitions for the cut-off frequency of the turbulence spectrum.

Apart from these three quantitative properties, an analytic model is applied to qualitatively model the shape of the lidar
measurement spectrum and its difference with respect to a reference, in this case the hot wire measurement spectrum.





## 2 Methodology, Part I: The Measurement Equipment

This section introduces the measurement infrastructure and equipment, which consists of the wind tunnel with an active grid, two short-range WindScanners and a single, mean wind component hot wire anemometer.

### 2.1 The Turbulent Wind Tunnel

ForWind – University of Oldenburg operates a Göttinger type open jet wind tunnel with a 30 m long test section and a 3 m by 3 m outlet, as shown in Fig. 1. The wind tunnel enables wind speeds up to 32 m s$^{-1}$ and has a turbulence intensity of approximately 0.3% in the case of an undisturbed flow. More details are described by Kröger et al. (2018). At the nozzle, either a passive or an active grid can be mounted. In Fig. 1 the nozzle is equipped with an active grid, consisting of 80 individually controllable shafts, to which 840 flaps are mounted. The flow through the outlet is regulated by dynamically varying the

inclination angle of the flaps, which is related to the local blockage of the flow. The active grid can be used in a passive configuration, where the user fixes the flap angles on all shafts, in order to create a steady flow with a low turbulence intensity and fixed shear layer. However, when the active grid is used to run user-defined protocols, it is also possible to generate higher turbulence intensities, modelled turbulence and reproduce prescribed wind speed time series, e.g. derived from lidar measurements of the wind in the open field (Kröger et al., 2018; Neuhaus et al., 2020, 2021). This enables the generation of

reproducible wind conditions for various purposes, e.g. wind turbine model tests (Petrović et al., 2019).

Five six-metre long test sections are available in the wind tunnel to build a closed measurement section. However, for the measurement campaign described in this paper, they are placed near the walls of the wind tunnel. Three of them can be seen on the right side of the nozzle in Fig. 1. The two remaining ones are parked at the back of the wind tunnel and serve as measurement platforms for the lidars, as illustrated by Fig. 2. In both figures, a red frame is drawn to indicate the nozzle with

the active grid.

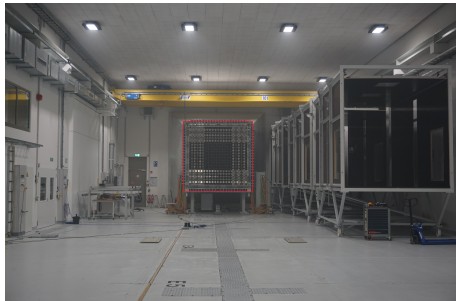

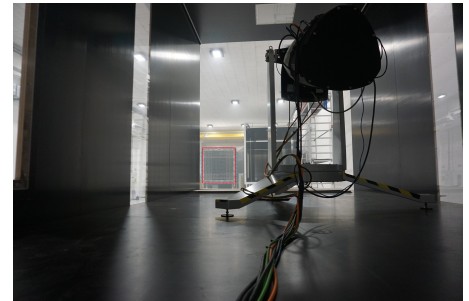

**Figure 1.** The open jet wind tunnel including the active grid at the nozzle.

**Figure 2.** View into the wind tunnel from the perspective of Wind-Scanner 1.

### 2.2 The Short-Range WindScanner

The lidars used for the measurements are continuous-wave short-range WindScanners with a 6" aperture (Mikkelsen, 2014; Mikkelsen et al., 2017) developed and manufactured by the Technical University of Denmark (see Fig. 3). The wind speed





measured by the WindScanner is based on coherent detection and is an absolute wind speed measurement, determined solely
by the laser's wave length and the Doppler shift in Hertz. Opposed to hot wire anemometers, Doppler wind lidars do not
need calibration. A recent paper by Pedersen and Courtney (2021) has shown that the WindScanner measured wind speeds
are represented with less than 0.1% absolute uncertainty. The WindScanners set their measurement range by adjusting their
focus setting of the telescopes. By changing the metal rods that attach the focus stages to the lens, different focus ranges, hence

different scan regimes can be achieved. The short-range mode allows the lidars to scan and measure at distances between 20 m
and 300 m, whereas the extra short-range mode, used for the measurement campaign described in this paper, decreases the
focus range to an interval between 12 m and 37 m. An important aspect of this reduction in focus length is the corresponding
smaller probe length (see Sect. 3.1). Both features are beneficial for wind tunnel measurements.

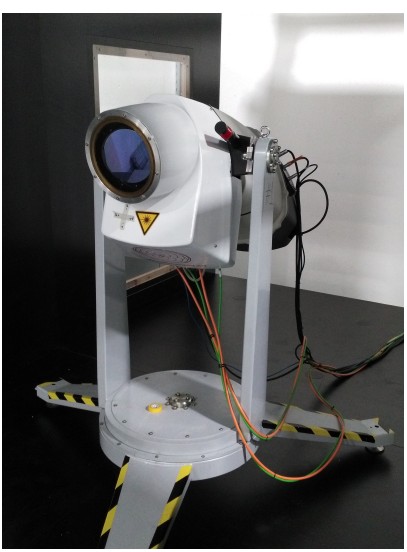

**Figure 3.** One of the two short-range WindScanners in the wind tunnel.

The two WindScanners are temporally synchronised at their maximum sampling rate of 451.7 Hz. Because the lidars have

a steerable scan head that allows the beam to point into user-defined pointing directions within a cone with a 120° opening
angle, they are able to scan any desired trajectory. The pointing accuracy of the two lidars for a staring mode measurement was
determined to be below 1 cm for the wind tunnel measurement campaign described here, which was guaranteed by a precise
measurement setup geometry and a manual confirmation and adjustment of the focus point of both WindScanners. In order to
maintain a sufficient amount of aerosols in the wind tunnel to reflect the laser beam, seeding was applied every few hours.

## 2.3    The Hot Wire Anemometer

The hot wire anemometer system used as reference for the wind tunnel measurement campaign consisted of a 54N80 Multi-
Channel CTA system in combination with a MiniCTA hot wire probe, both acquired from the manufacturer Dantec Dynamics.
The sensor is a basic, one-dimensional hot wire probe with a wire thickness of 5 μm. It was mounted on a structure with an





extended rod and aligned with the mean flow direction, i.e. the $x$-direction, of the wind tunnel. It was calibrated manually every
day, before and after the measurement, whilst taking into account a temperature correction. Figure 4 shows the hot wire probe installation in the wind tunnel, together with the manually adjusted focus points of each respective WindScanner laser beam, visualised by an infra-red detector card.

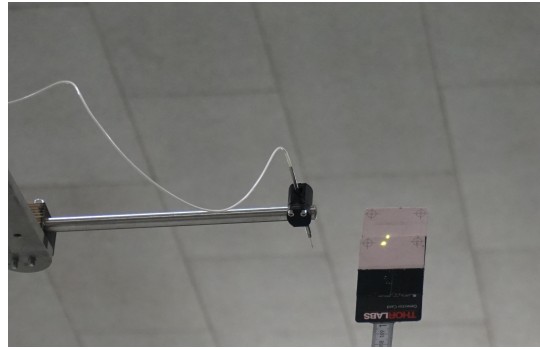

**Figure 4.** Setup of the hot wire system next to the adjusted focus points of the WindScanners' laser beams.

## 2.4   The Experimental Setup

For an optimal setup of the WindScanner system in the wind tunnel, three important aspects have to be considered:

1. For the largest possible measurement domain coverage within the wind tunnel, with particular interest in the region directly in front of the active grid, and taking into account the minimum focus distance of 12 m, the WindScanners have to be placed as far downstream in the wind tunnel as possible. This also eliminates the obstruction of the flow by the lidars.

     2. In order to accurately calculate a two-dimensional wind speed vector, there has to be a favourable opening angle between
110        the beams of the WindScanners (Stawiarski et al., 2013), preferably larger than $30°$. This means that the transverse distance between the WindScanners has to be maximised.

     3. The lidars should measure with a near-zero elevation angle to avoid disturbance by the vertical wind speed component of the turbulent flow.

Although these three aspects cannot be satisfied in an optimal way simultaneously, an acceptable compromise between them
is sketched in Fig. 5. The flow enters the measurement section from the left, where the active grid is positioned at the wind tunnel nozzle. WindScanners 1 and 2 are installed at distances of 27.48 m and 27.32 m downstream of the nozzle, respectively. Therefore they can cover the region between the active grid and approximately $x = 15.5$ m downstream. The lidars are installed at maximum transverse distance, inside empty test sections that were parked at the back end of the wind tunnel, such that their separation along the $y$-axis has the largest possible value of 9.7 m. The hot wire probe (see Fig. 4) was installed at the position
indicated in Table 1. It was calibrated twice every day, both at the start and at the end of the measurement. The lidars were





focused at a point in space 7 cm away from the hot wire in the positive $y$-direction, to avoid any mutual interference between the sensors (see Fig. 4). The opening angle between the lidars' laser beams for this point is 28.4°, which is acceptable for a two-dimensional wind speed reconstruction (Stawiarski et al., 2013).

**Table 1.** Coordinates relevant to the experimental setup.

|  | $x$ [m] | $y$ [m] | $z$ [m] |
|---|---|---|---|
| Active grid centre | 0.00 | 0.00 | 3.00 |
| WindScanner 1 | 27.48 | 4.74 | 2.86 |
| WindScanner 2 | 27.32 | −4.92 | 2.88 |
| Hot wire | 8.35 | 0.21 | 2.99 |
| Lidar focus point | 8.35 | 0.28 | 2.96 |

**Table 2.** Setup parameters describing the WindScanner staring mode scan.

| Parameter | WindScanner 1 | WindScanner 2 |
|---|---|---|
| $\chi$ [°] | 12.8 | −15.1 |
| $\delta$ [°] | 0.3 | 0.3 |
| $\beta$ [°] | 12.8 | 15.1 |
| $L$ [cm] | 13.0 | 13.9 |

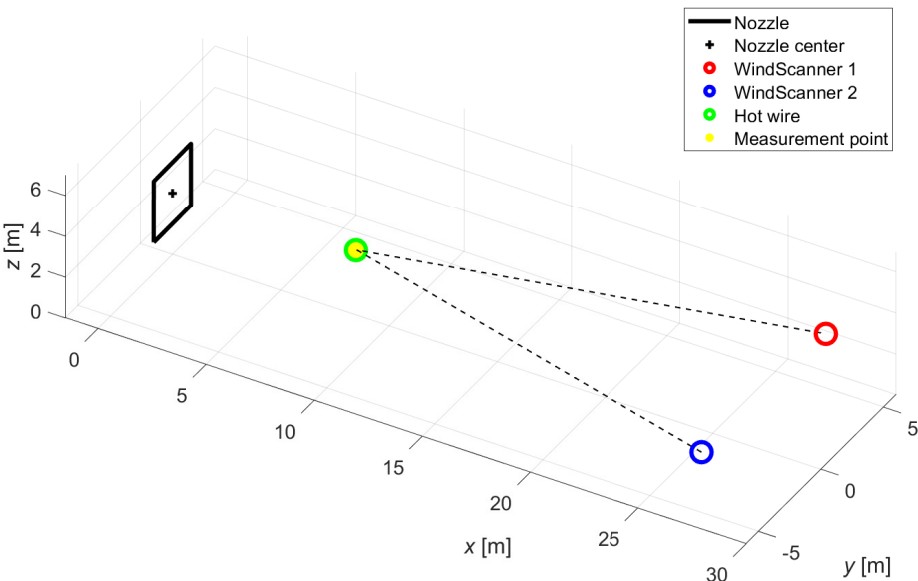

**Figure 5.** Schematic measurement setup of the two WindScanner lidars and the hot wire in the wind tunnel.

The temporal synchronisation between the WindScanners and the hot wire was established by performing a cross-correlation between the smoothed time series at 451.7 Hz sampling rate.

## 3 Methodology, Part II: The Physical Models

This section introduces the theory used to physically describe and model the lidar measurement principle. A more in-depth description of lidar theory is provided by van Dooren (2021).





## 3.1 The Lidar Measurement Principle

One of the most important properties of a coherent detection wind lidar is that the measurement region is not an infinitesimal point in space, but a probe volume, which is usually a thin cylinder with a radius of less than a millimetre at short ranges and a length in the order of centimetres or metres, depending on the focus distance. A simple and commonly used physical model for the probe length averaging effect of a lidar measurement comprises a low-pass filter with the shape of a Lorentzian function (Angelou et al., 2012; Slinger and Harris, 2012). In the following, this model is implemented and compared with the actual measurement. This is relevant for the analysis of the measurements in both the wave number space and the frequency domain.

The Lorentzian probe length function is represented by Eq. (1);

$$F = \frac{1}{\pi} \frac{\frac{1}{2}L}{\left(\frac{1}{2}L\right)^2 + (s - d_f)^2},$$
(1)

where $F$ is the normalised intensity of transmitted laser power along the line-of-sight direction coordinate $s$, resembling the distance from the lens, $d_f$ is the focus distance, and $L$ is the probe length of the WindScanner defined by Eq. (2);

$$L = 2 \frac{\lambda d_f^2}{\pi a^2},$$
(2)

where $\lambda$ is the laser wavelength of 1.55 μm, $a$ is the effective radius of 56 mm of the lidar's 6" aperture telescope used for emitting and receiving the laser signal and $L$ is the full width half maximum of the Lorentzian intensity profile symmetrically centred about the focus point, also known as the probe length. As the backscattered radiation is proportional to the focused laser beam's power intensity, most of the laser light will be reflected by aerosols in this volume, however, a smaller backscatter contribution from outside these bounds is unavoidable.

For the aforementioned extra short-range focus distance range, the probe length varies between 4.5 cm and 43.1 cm. A plot indicating the probe length for the entire focus distance envelope can be consulted in Fig. 6.

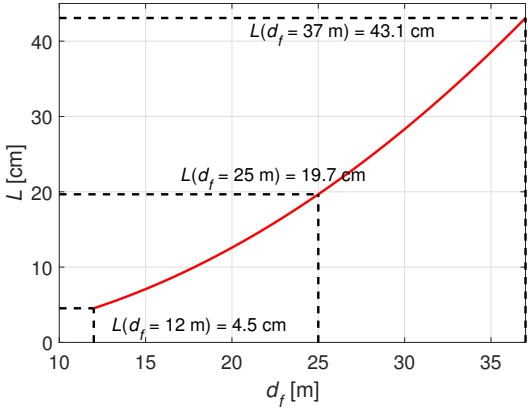

**Figure 6.** Plot of the theoretical probe length $L$ of the 6" aperture WindScanner for the applied focus distance $d_f$ ranging from 12 m to 37 m.





A single Doppler wind lidar can only measure a one-dimensional component of the wind speed, i.e. the projection of the full three-dimensional wind velocity vector onto its line-of-sight direction, indicated by Eq. (3);

$$v_{\mathrm{LOS}} = \begin{bmatrix} \cos(\chi)\cos(\delta) \\ \sin(\chi)\cos(\delta) \\ \sin(\delta) \end{bmatrix} \cdot \begin{bmatrix} u \\ v \\ w \end{bmatrix},$$

(3)

where $v_{\mathrm{LOS}}$ is the line-of-sight wind speed measured by the lidar, $\chi$ and $\delta$ are the azimuth and elevation angles of the line-of-sight direction, respectively, and $u$, $v$ and $w$ are the wind speed components in $x$-, $y$- and $z$-direction, respectively.

Assuming the vertical wind speed $w$ to be zero, a two-dimensional horizontal wind vector is calculated from two linearly independent measurements from different lidars through Eq. (4);

$$\begin{bmatrix} \cos(\chi_1)\cos(\delta_1) & \sin(\chi_1)\cos(\delta_1) \\ \cos(\chi_2)\cos(\delta_2) & \sin(\chi_2)\cos(\delta_2) \end{bmatrix} \begin{bmatrix} u \\ v \end{bmatrix} = \begin{bmatrix} v_{\mathrm{LOS}_1} \\ v_{\mathrm{LOS}_2} \end{bmatrix},$$

(4)

where the indices 1 and 2 refer to the respective lidar. The wind speed vector $[u\,v]^{\mathsf{T}}$ can be calculated as the solution to this linear system, which we will refer to as dual-Doppler reconstruction.

Apart from the dual-Doppler reconstruction, for which two lidars are needed, it is also possible to calculate each single-lidar wind speed estimate. To be able to compare the performance of the two WindScanners, most of the results in this paper are based on one-dimensional projection measurements, calculated by Eq. (5);

$$u_p = \frac{v_{\mathrm{LOS}}}{\cos(\chi)\cos(\delta)},$$

(5)

where $u_p$ is the projected $u$-component, under the assumption that $v$ and $w$ are both zero.

### 3.2  Lidar Spectral Modelling

To understand and express the measurement principle of cw-lidar better, the measurement can be further investigated in the frequency domain (Angelou et al., 2012; Held and Mann, 2018). Our aim is to define a model for the spectral density function of a lidar measurement, encompassing a theoretical probe volume averaging filter and a white noise term. The spatial filter which the lidar's probe volume exerts on the turbulence along its line-of-sight direction is described by the aforementioned Lorentzian function in Eq. (1). This probe volume acts as a low-pass filter that reduces the spectral energy content of the turbulent eddies with wave lengths smaller than the axial extent of the probe volume, however, the spectrum measured by the lidar usually still shows a significant energy content in the region above that cut-off frequency. The lidar measured spectrum may be artificially increased for higher frequencies due to random noise on the measurements, associated with low signal-to-noise levels, which depend on the aerosol density in the wind tunnel. This is in agreement with previous research (Sjöholm et al., 2009; Angelou et al., 2012). Therefore, the model also includes a white noise term.

The proposed model in the time domain is given by Eq. (6);

$$\hat{v}_{\mathrm{LOS}} = \frac{1}{\pi} \int_{-\infty}^{\infty} \frac{\frac{1}{2}L}{\left(\frac{1}{2}L\right)^2 + (s - d_f)^2} u(s)\,ds + \sigma_\eta \eta,$$

(6)





where $\hat{v}_{\mathrm{LOS}}$ is the modelled WindScanner line-of-sight time series, $u(s)$ is the actual wind speed $u\cos(\chi)\cos(\delta)$ to which the Lorentzian filter is applied, $s$ is the coordinate along the line-of-sight direction, $\eta$ is a dimensionless Gaussian noise term with a mean of zero and a standard deviation of 1, and $\sigma_\eta$ is the magnitude of the standard deviation (in m s$^{-1}$) of the noise term. Please note that the estimated $\hat{v}_{\mathrm{LOS}}$ is not meant to closely correlate to the actual $v_{\mathrm{LOS}}$ measurement in the time domain, which is impossible due to the random nature of the modelled noise term. The goal of this approach is to find a time series $\hat{v}_{\mathrm{LOS}}$ with the same features in the frequency domain as $v_{\mathrm{LOS}}$, by qualitatively comparing the spectral density functions.

A vital assumption for the model in Eq. (6) is that the hot wire measurements, although also low-pass filtered, but at a much higher frequency, resemble the actual wind speed at the focus point of the lidar. In order to apply the spatial Lorentzian filter to a temporally sampled time series, we apply Taylor's frozen turbulence hypothesis (Taylor, 1938). This relies on the assumption that turbulence is advected through the lidar's probe volume by the mean wind speed without decaying.

In the frequency domain, the Lorentzian filter can be expressed in terms of the spectral transfer function $T(\beta, f)$ in Eq. (6);

$$T(\beta,f) = \frac{11}{9\sqrt{\pi}}\frac{\Gamma(1/3)}{\Gamma(5/6)}\sin^{5/3}(\beta)\left(\pi\frac{f}{f_c}\right)^{-1} + \frac{1}{2}(7\cos(2\beta)-5)e^{-\pi\frac{f}{f_c}}, \tag{7}$$

where the angle $\beta$ denotes the angle between the lidar's line-of-sight direction and the mean flow direction, $f$ is frequency, $f_c$ is the probe length cut-off frequency to be defined in Sect. 4.1 and $\Gamma$ is the Gamma function. This function can be applied as a multiplier on an un-truncated turbulent inertial sub-range spectrum with an exponent equal to $-5/3$ according to Komolgorov (1941) of the wind component in the mean wind direction, in order to resemble the spectrum of a time series measured by the lidar, pointed at the mean wind direction under an angle $\beta$ that has passed through a Lorentzian filter. The full derivation of Eq. (7) is provided in Appendix A.

Since the mean wind tunnel flow is aligned with the $x$-direction, we can express the angle $\beta$ as a combination of both azimuth and elevation angles of the WindScanners, as stated by Eq. (8);

$$\beta = \arccos\big(\cos(\chi)\cos(\delta)\big). \tag{8}$$

In case of zero misalignment between the lidar line-of-sight direction and the mean flow direction, the spectral transfer function $T(f)$ reduces to:

$$T(f) = e^{-\pi\frac{f}{f_c}}. \tag{9}$$

We are now able to model the observed spectral density function of a lidar measuring turbulence within the inertial sub-range, simply by multiplying the observed hot wire spectrum, which represents the un-filtered inertial sub-range spectrum of the turbulence component aligned with the mean wind direction, with the lidar transfer function $T(\beta, f)$, and subsequently adding a white noise term $S_\eta(f)$ to it, as Eq. (10) illustrates;

$$S(f) = T(\beta,f)S_u(f) + \sigma_\eta^2 S_\eta(f), \tag{10}$$

where $S(f)$ is the modelled lidar spectrum, $S_u(f)$ is the hot wire spectrum, and $S_\eta(f)$ is the spectrum of white noise with a mean of zero and a standard deviation of 1, which has to be scaled with the variance $\sigma_\eta^2$ for each specific case. White noise





(Kuo, 1996) has the property that its power density spectrum $S_\eta(f)$ shows the same energy content at each given frequency, which lets us maintain a low complexity of the model. Although the spectral density function is proportional to the square of a Fourier Transform of the measured time series, the spectral densities of both the filtered hot wire time series and the white noise can be joined by a simple addition, since they are assumed to be uncorrelated.

To complete the model, the standard deviation of the noise $\sigma_\eta$ has to be expressed. We hypothesised that this variable is connected to the energy dissipation rate $\varepsilon$ of the classic Komolgorov spectrum, the mean wind speed $\mu_u$ or a combination of both. Based on dimensional analysis, investigation of the linear dependency of $\sigma_\eta$ on various combinations of the aforementioned variables, and manual tuning of all available data, the following model is proposed;

$$\hat{\sigma}_\eta = C\sqrt{\frac{\mu_u \varepsilon}{g}},\tag{11}$$

where $C$ is a constant of 0.132 found by performing a linear fit between calculated and manually tuned $\sigma_\eta$ values, $\varepsilon$ is the energy dissipation rate calculated by fitting the theoretical Komolgorov spectrum in Eq. (12) adapted from Wang et al. (2020) to the hot wire spectrum in the inertial sub-range, and $g$ is the gravitational acceleration constant equal to 9.81 m s$^{-2}$.

$$S(f) = 0.5\varepsilon^{\frac{2}{3}}\left(\frac{2\pi f}{\mu_u}\right)^{-\frac{5}{3}}\tag{12}$$

## 4 Results and Discussion

The results and discussion section is divided into three parts: First the WindScanner lidar measurements are validated by comparing them to hot wire anemometer measurements. Afterwards the lidars' ability to measure turbulent gusts generated by the active grid in the wind tunnel is analysed. Finally the power spectral density model for continuous-wave lidar is evaluated.

### 4.1 Comparison between WindScanner and Hot Wire Anemometer Measurements

As an evaluation of the equipment and the measurement setup, a comparison of the WindScanner lidar meaurements to those obtained from a hot wire anemometer was carried out on a 10-minute sampling time basis, for cases of both low and high turbulence intensity. The WindScanner and hot wire anemometer measurements were sampled at 451.7 Hz and 1000 Hz, respectively. The latter was down-sampled to the lidar sampling rate by means of linear interpolation.





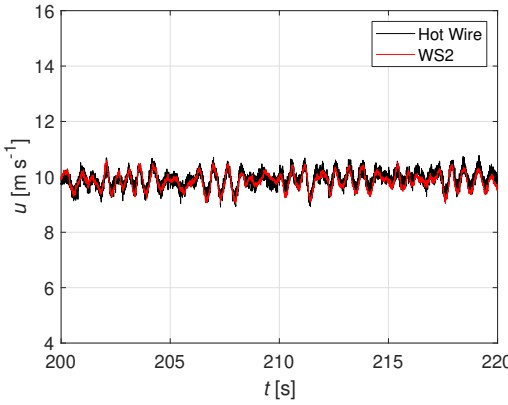

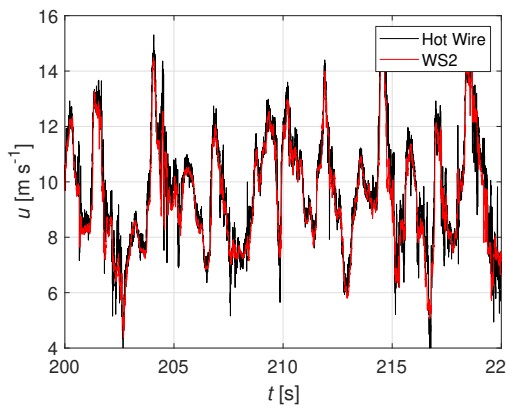

**Figure 7.** Snapshot of the 10-minute time series measured by Wind-Scanner 2 and the hot wire anemometer for a turbulence intensity of 3% (Case 1a).

**Figure 8.** Snapshot of the 10-minute time series measured by Wind-Scanner 2 and the hot wire anemometer for a turbulence intensity of 22% (Case 1b).

Figures 7 and 8 show a 20-second snapshot of the 10-minute time series measured by the hot wire and WindScanner 2, for two different turbulent conditions; Cases 1a and 1b, respectively. The black line resembles the wind speed measured by the hot wire, which is already aligned with the $x$-direction to measure the $u$-component. The red line represents the projected $u_p$-component of the wind speed, calculated by applying Eq. (5) to the measured line-of-sight wind speed of WindScanner 2.

Case 1a with TI = 3% was established by a fully opened active grid, effectively operating in a passive mode. For Case 1b with TI = 22% the active grid ran a turbulent protocol based on the approach by Neuhaus et al. (2020). An overview of the corresponding statistics is provided in Table 3, where $\mu_u$ and $\sigma_u$ are the mean and the standard deviation of the wind speed component $u$, respectively. The aforementioned turbulence intensity TI is defined on the basis of those statistics by Eq. (13):

$$\text{TI} = \frac{\sigma_u}{\mu_u}. \tag{13}$$

By looking at the time series themselves, some differences in the turbulence behaviour can already be observed. For Case 1a there is a visible dominating frequency, with additional superimposed small-scale fluctuations. In Case 1b we do not identify such a distinct dominating frequency, but instead see a superposition of relatively large-scale structures and some small-scale fluctuations.

**Table 3.** Statistics of the 10-minute hot wire and WindScanner time series of the $u$-component of the wind.

|  |  | Hot wire | WindScanner 1 | WindScanner 2 |
|---|---|---|---|---|
| Case 1a with TI = 3% | $\mu_u$ [m s$^{-1}$] | 9.89 | 9.79 | 9.77 |
|  | $\sigma_u$ [m s$^{-1}$] | 0.31 | 0.30 | 0.31 |
| Case 1b with TI = 22% | $\mu_u$ [m s$^{-1}$] | 10.11 | 9.88 | 9.85 |
|  | $\sigma_u$ [m s$^{-1}$] | 2.26 | 2.30 | 2.23 |

The values of the mean $u$-component in Table 3 indicate that both WindScanners have a negative bias of around 0.1 m s$^{-1}$

245 for Case 1a and around 0.25 m s$^{-1}$ for Case 1b relative to the hot wire measurement. This effect might be related to the measurement volume, which has a finite probe length $L$ and is not aligned with the $x$-direction. Even though the wind speed has been corrected for the scanning angles $\chi$ and $\delta$ by means of Eq. (5), the way the probe lengths of about 13.0 cm and 13.9 cm respectively (see Table 2) cross through the measurement point causes an averaging of the $u$-component with contributions from outside the desired point. The bias may also be related to the uncertainty of the hot wire calibration, given that the

250 WindScanners proved to have an uncertainty below 0.1% (Pedersen and Courtney, 2021). In contrast to the wind speed mean, only minor differences can be observed between the measured standard deviations.

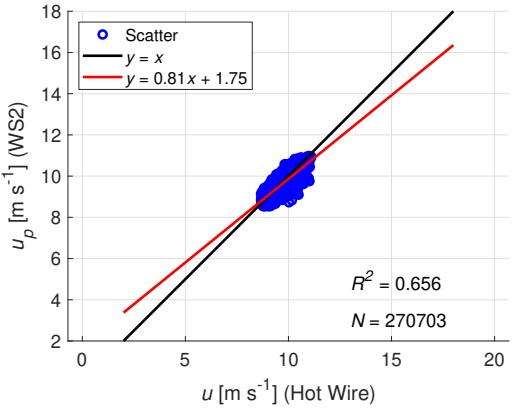
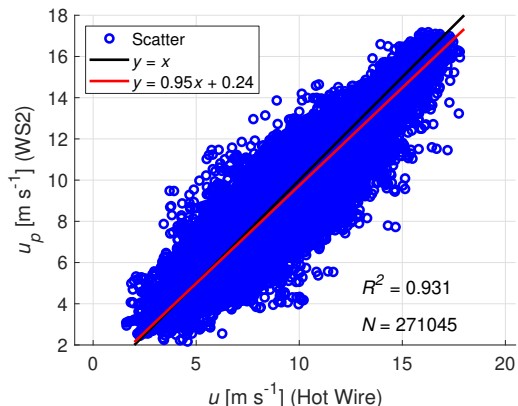

**Figure 9.** Correlation between WindScanner 2 and the hot wire anemometer for a turbulence intensity of 3% (Case 1a).

**Figure 10.** Correlation between WindScanner 2 and the hot wire anemometer for a turbulence intensity of 22% (Case 1b).

Figures 9 and 10 show the linear regression plots of the correlation of the full time series corresponding to Figs. 7 and 8, respectively. For Cases 1a and 1b, the goodness of fit coefficients $R^2$ are 0.656 and 0.931, respectively. It seems counter-

255 intuitive that the case with higher turbulence is measured more accurately by the WindScanner. However, it should be kept in mind that the spectral characteristics of both time series strongly differ. An explanation will be given in the following after performing a spectral analysis.

Figures 11 and 12 display the spectral density functions of the wind speed measurements corresponding to Figs. 9 and 10, respectively. To create the spectra, and all other spectra in the following, we performed Hann smoothing with $N = 10$ non-

260 overlapping Hann windows. Because of the lidar measurement principle, turbulent structures with a length scale smaller than the probe length are partly filtered out. Assuming Taylor's frozen turbulence hypothesis and the Nyquist criterion, the probe length cut-off frequency $f_c$ corresponding to the smallest turbulence scales that can be measured relatively un-truncated by the lidars is estimated with Eq. (14);

$$f_c = \frac{1}{2} \frac{u_\infty}{L}, \tag{14}$$

265 where $L$ is the probe length and $u_\infty$ is the free-stream wind speed in the wind tunnel, assumed to be represented by the average of the wind speed time series measured by the hot wire anemometer. For Case 1a the probe length of WindScanner 2 is





13.9 cm and the mean wind speed is 9.89 m s$^{-1}$, leading to a probe length cut-off frequency of 35.6 Hz. For Case 1b, the mean wind speed increased to 10.11 m s$^{-1}$, leading to a slightly higher cut-off frequency of 36.4 Hz. These values are indicated in Figs. 11 and 12 with a magenta line.

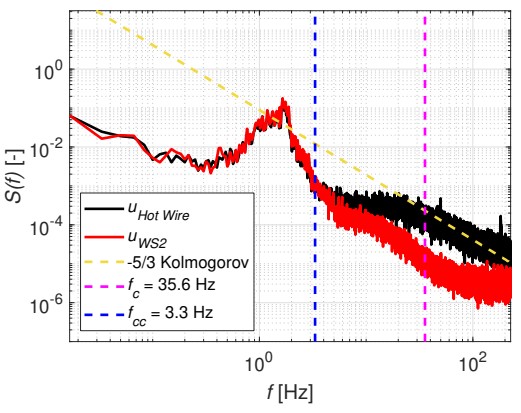

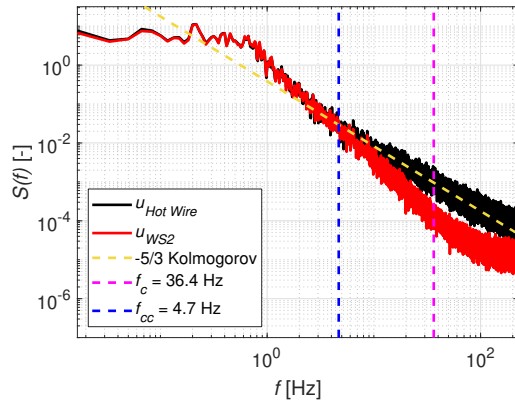

**Figure 11.** Spectrum of WindScanner 2 and the hot wire anemometer for a turbulence intensity of 3% (Case 1a).

**Figure 12.** Spectrum of WindScanner 2 and the hot wire anemometer for a turbulence intensity of 22% (Case 1b).

By means of these spectra, we can now readdress the observation regarding the time series in Figs. 7 and 8 and the difference in the goodness of fit coefficients of the linear regression plots in Figs. 9 and 10. The spectrum for the low turbulence case shows a clear signature of high energy content around 1.65 Hz, which confirms our observation of a dominating frequency based on visual inspection of the time series. This frequency has been addressed as a common phenomenon in open duct wind tunnels before (Wickern et al., 2000), and can be explained by vortex shedding occurring around the nozzle. This phenomenon causes a pulsating pressure wave, leaving a characteristic fluctuation pattern in the wind speed time series in Fig. 7. According to Wickern et al. (2000), the vortex shedding frequency of an open duct can be calculated by Eq. (15);

$$f_{\text{vs}} = \text{St} \frac{u_\infty}{D}, \tag{15}$$

where $f_{\text{vs}}$ is the vortex shedding frequency, $\text{St}$ is the Strouhal number and $D$ is the diameter of the nozzle, which is equated here with the square width $W$ of the nozzle. Solving this equation for our known parameters $u_\infty = 9.89$ m s$^{-1}$ and $W = 3$ m a Strouhal number $\text{St} = 0.5$ is found, which is in accordance with the findings by Ahuja et al. (1997) and Wickern et al. (2000).

The high TI spectrum does not show this peak, but has a significantly higher energy content broadly distributed over the lower frequency region, representing the more dominant presence of longer length scales. This explains why the WindScanners have a better correlation with the hot wire for the high turbulence case, since the cw-lidars are much better capable of measuring large-scale structures over small-scale fluctuations. The small scales play a more dominant role in the spectrum in the low TI case, which is highlighted by the discrepancy between the WindScanner and hot wire spectra above around 5 Hz and 10 Hz, for the low and high TI cases respectively. These observed frequencies at which the WindScanner spectra drop from the hot wire spectra imply a significantly lower limit than the prediction by Eq. (14) according to Taylor's frozen turbulence hypothesis





and the Nyquist criterion. A possible reason is the misalignment of the probe volumes with the $x$-axis, which violates the
one-dimensionality assumed in order to apply this principle on the single $u$-component.

### 4.2   Analysis of WindScanner Measurements of Turbulent Gusts

In order to analyse the WindScanners' ability to measure gusts, we set up an additional five cases, where an identical 10-minute
gust protocol based on the approach by Neuhaus et al. (2021) was run by the active grid, and only the mean wind speed in the
wind tunnel was varied between 2.36 m s$^{-1}$ and 11.25 m s$^{-1}$. However, the varying wind speed naturally affects the coefficient
of variation $c_\mathrm{v} = \sigma_u/\mu_u$, i.e. between 11.7% and 16.4%. This is illustrated by Fig. 13, which serves as a summary of the hot
wire statistics for the five investigated cases. This particular active grid protocol was selected for its ability to generate a time
series containing structures with varying length scales, providing a suitable test case for the WindScanner measurement. Please
note that we opt for the coefficient of variation $c_\mathrm{v}$ instead of the turbulence intensity TI here, since the fluctuations introduced
by the artificially generated gusts do not comply to the classic definition of TI.

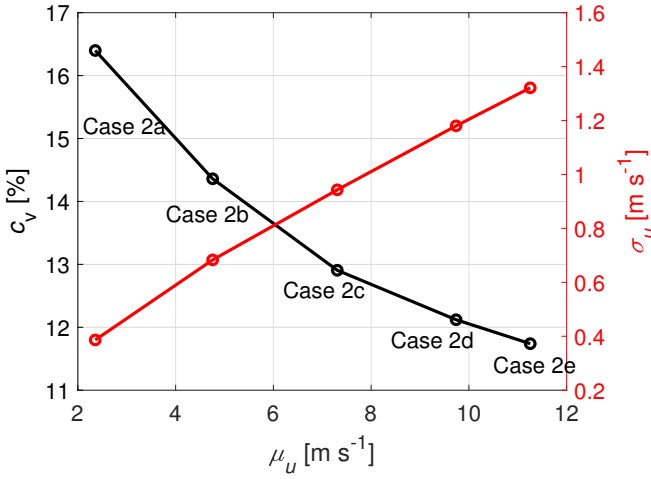

**Figure 13.** Plot of the standard deviation $\sigma_u$ and coefficient of variation $c_\mathrm{v}$ as a function of the mean wind speed $\mu_u$ calculated from the hot
wire time series of each of the five cases 2a–2e.

Even though the hot wire was calibrated twice every day, the post-processing revealed a bias relative to the WindScanner
measurements equal to $-5.2\%$ of the mean wind speed for this data. This bias was constant throughout Cases 2a–2e of the
measurement campaign, i.e. the measurement of turbulent gusts, but did not occur at all during the validation measurement
(Cases 1a and 1b) that was done a few months prior. After careful reassessment of the entire measurement chain, we assumed it
was caused by a faulty hot wire calibration, and therefore we corrected the hot wire measurements by a factor of 0.948 as a part
of the post-processing to yield a satisfactory match between the measurement sensors. This has to be taken into consideration
when assessing the results.





For a more comprehensive overview of the statistics measured during each of the five cases, see Table 4. It contains the mean and standard deviation values of the 10-minute time series measured by the hot wire, WindScanner 1 and 2, as well as the dual-Doppler reconstructed wind speed.

**Table 4.** Statistics of the hot wire and WindScanner time series of the $u$-component of the wind speed.

|  | Variable | Hot wire | WindScanner 1 | WindScanner 2 | Dual-Doppler |
|---|---|---|---|---|---|
| Case 2a | $\mu_u$ [m s$^{-1}$] | 2.36 | 2.32 | 2.32 | 2.32 |
|  | $\sigma_u$ [m s$^{-1}$] | 0.39 | 0.36 | 0.35 | 0.35 |
|  | $c_\mathrm{v}$ [%] | 16.4 |  |  |  |
| Case 2b | $\mu_u$ [m s$^{-1}$] | 4.76 | 4.77 | 4.72 | 4.75 |
|  | $\sigma_u$ [m s$^{-1}$] | 0.68 | 0.69 | 0.68 | 0.68 |
|  | $c_\mathrm{v}$ [%] | 14.4 |  |  |  |
| Case 2c | $\mu_u$ [m s$^{-1}$] | 7.31 | 7.31 | 7.24 | 7.28 |
|  | $\sigma_u$ [m s$^{-1}$] | 0.94 | 0.96 | 0.94 | 0.95 |
|  | $c_\mathrm{v}$ [%] | 12.9 |  |  |  |
| Case 2d | $\mu_u$ [m s$^{-1}$] | 9.74 | 9.74 | 9.65 | 9.70 |
|  | $\sigma_u$ [m s$^{-1}$] | 1.18 | 1.21 | 1.20 | 1.20 |
|  | $c_\mathrm{v}$ [%] | 12.1 |  |  |  |
| Case 2e | $\mu_u$ [m s$^{-1}$] | 11.25 | 11.29 | 11.18 | 11.24 |
|  | $\sigma_u$ [m s$^{-1}$] | 1.32 | 1.38 | 1.36 | 1.37 |
|  | $c_\mathrm{v}$ [%] | 11.7 |  |  |  |

Figure 14 shows the measured wind speed time series, produced by the active grid running a gust protocol, for Case 2c. The protocol cycles through a series of gusts, facilitating a combination of gusts with varying duration and intensity. This allows the analysis the performance of the cw-lidar for a wide range of turbulence scales. The protocol has a duration of approximately 100 s, which is repeated six times to yield a 10-minute time series. Each of the cycles contains ten distinct gusts. Figure 15 displays three of such gusts, which occur between $t = 200$ s and $t = 220$ s.




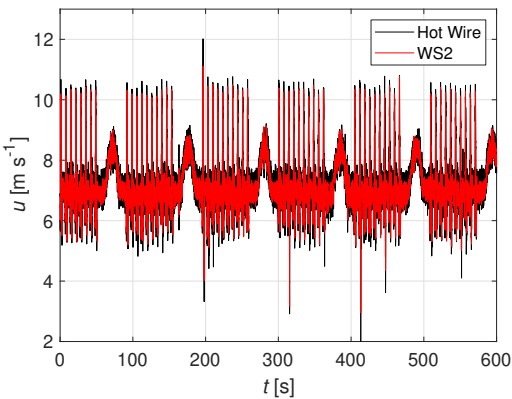
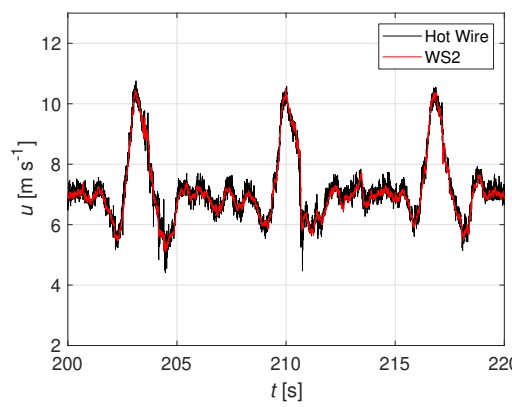

**Figure 14.** The 10-minute time series measured by WindScanner 2 and the hot wire anemometer for a coefficient of variation of 12.9% (Case 2c).

**Figure 15.** Snapshot of the 10-minute time series measured by WindScanner 2 and the hot wire anemometer for a coefficient of variation of 12.9% (Case 2c).

In both Figs. 14 and 15, the black line is the wind speed $u$ measured by the hot wire and the red line resembles the projected $u_p$-component of the wind speed of WindScanner 2. At first glance, it can be seen that the WindScanner measurements have a narrower spread than the hot wire, indicating that smallest turbulence scales are filtered out in the probe volume.

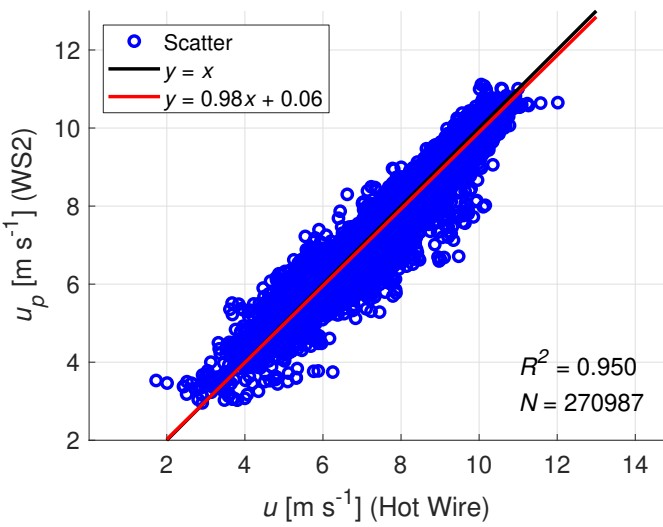

**Figure 16.** Correlation between WindScanner 2 and the hot wire anemometer for a coefficient of variation of 12.9% (Case 2c).

Figure 16 illustrates the correlation between the projected wind speed based on WindScanner 2 and the hot wire measurement for Case 2c, corresponding to the time series of Fig. 14 with 451.7 Hz resolution. It shows a goodness of fit coefficient of $R^2 = 0.950$, which is similar to the value found for Case 1b in Fig. 10. The linear regression curve has a slope of 0.98 and an offset of 0.06 m s$^{-1}$.



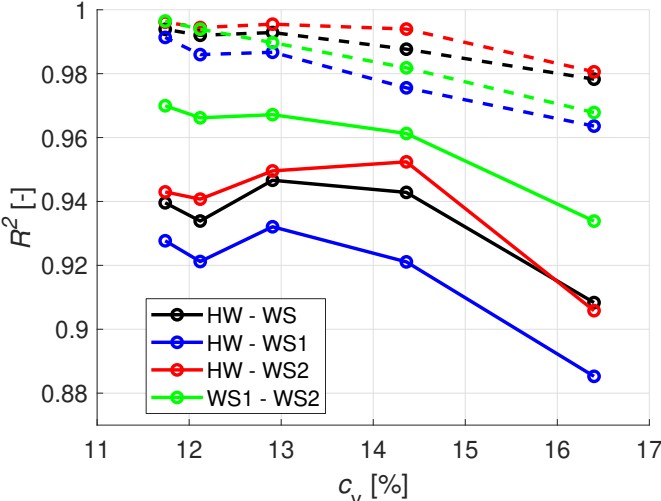

**Figure 17.** Correlation between WindScanner and hot wire time series, as a function of coefficient of variation, for both 451.7 Hz (solid lines) and 1 Hz (dashed lines) resolution. HW denotes the hot wire, WS1 and WS2 stand for WindScanner 1 and WindScanner 2, respectively, and WS resembles the dual-Doppler reconstruction.

To investigate the effect of coefficient of variation on the accuracy of the WindScanner measurement with respect to the hot wire, the correlation for Cases 2a–2e is plotted in Fig. 17. Here all solid lines represent the correlation of the highly resolved

451.7 Hz time series, and the respective dashed lines resemble the correlation between the 1 Hz averaged time series. The blue and red lines correspond to the correlation between the hot wire and WindScanners 1 and 2, respectively. The black curve refers to the dual-Doppler reconstructed wind speed. The green line resembles the correlation between both WindScanners.

We see a decreasing trend for the correlation of all sensors with respect to the coefficient of variation. However, for all cases the goodness of fit coefficient $R^2$ is higher than 0.88. For the highly resolved time series, WindScanner 2 correlates

better to the hot wire than WindScanner 1 does. The dual-Doppler reconstructed wind speed nearly overlaps with the curve for WindScanner 2. The correlation between the two WindScanners is the highest among all comparisons, which is not surprising considering they are identical devices. For all 1 Hz averaged cases the goodness of fit coefficient is above 0.96. The order in which the correlation curves are plotted is the same as for the highly resolved time series, the only exception being the correlation between the two WindScanners.





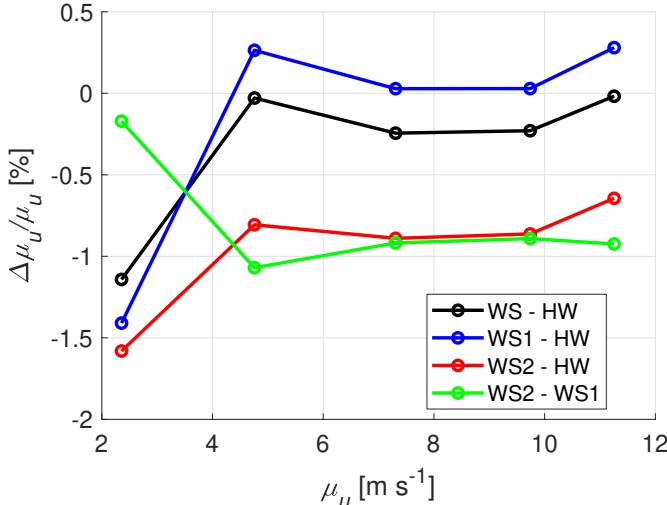

**Figure 18.** Plot of the relative mean wind speed difference with respect to the hot wire anemometer for each of the five cases at 451.7 Hz.

A second measure for the performance of the WindScanners besides the correlation analysis is the mean average error with respect to the hot wire, as illustrated by Fig. 18. The relative error $\Delta\mu_u/\mu_u$ is plotted as a function of the mean hot wire wind speed $\mu_u$, corresponding to the statistics in Table 4. Please consider that a hot wire calibration correction factor of 0.948 has been implemented here, as explained before in Sect. 2.4. The lowest errors are found for the dual-Doppler reconstructed wind speed, resembled by the black line. Here the error is down to $-1.1\%$ for the lowest wind speed. WindScanner 1 alone tends to

overestimate the wind speed by up to 0.3% (blue line), whereas WindScanner 2 tends to do the opposite, with an error down to $-1.6\%$ (red curve). There also seems to be a bias between WindScanner 1 and 2, which increases linearly with the mean wind speed. Different properties of the optical systems of the respective WindScanners are considered the most likely reason. Putting the relative errors in perspective to the findings of Pedersen and Courtney (2021), who found a 0.1% total calibration uncertainty when measuring the speed of a rotating flywheel, it is concluded that additional measurement uncertainties related

to the fluctuating wind speed and the atmospheric conditions inside the wind tunnel play an important role.

     Figure 19 presents the power spectrum of the time series corresponding to Case 2c. The black and red curves correspond to their respective time series in Figs. 14 and 15. The yellow line corresponds to Komolgorov turbulence decay in the inertial sub-range (Komolgorov, 1941). The vertical magenta line is the estimated probe length cut-off frequency defined by Eq. (14). Both sensors clearly indicate several peaks for frequencies lower than 1 Hz, which correspond to the periodicity in the gust protocol.

They do not have physical significance and have a much larger scale than the lowest scales detectable by the WindScanners. No differences can be distinguished between the two curves below approximately 3 Hz. Overall the slope of the hot wire spectrum is close to the $-5/3$ Komolgorov ratio. The spectrum of the WindScanner drops below the hot wire spectrum earlier than the predicted cut-off frequency, which we have seen before in Figs. 11 and 12. It is worth noticing that the slope of the WindScanner spectrum is not constant, but first has a large negative tendency and later bends back upwards to a nearly horizontal curve for

frequencies above 200 Hz. This phenomenon will be explained and modelled in Sect. 4.3.





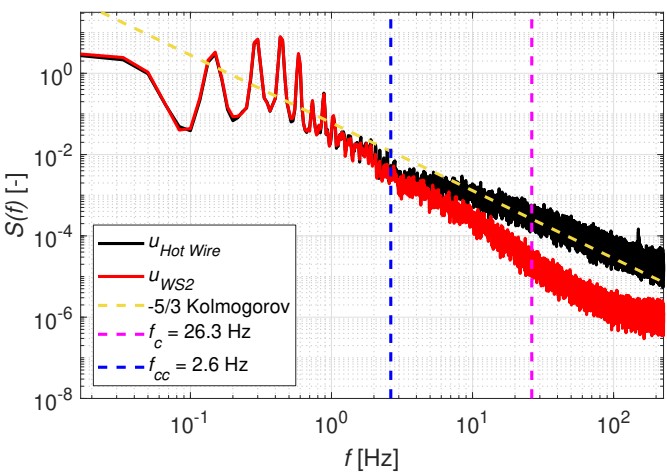

**Figure 19.** Spectrum of WindScanner 2 and the hot wire anemometer for a coefficient of variation of 12.9% (Case 2c).

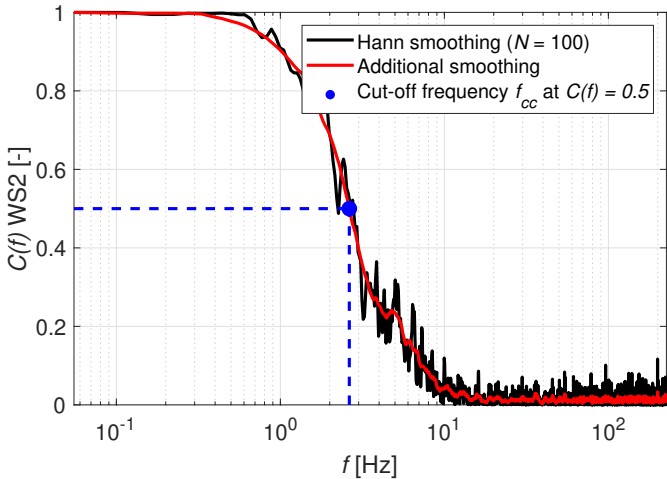

**Figure 20.** Coherence between WindScanner 2 and the hot wire anemometer for a coefficient of variation of 12.9% (Case 2c).

A different method, used to address the differences between the hot wire and cw-lidar time series in the frequency domain, is the coherence function. Figure 20 shows this curve for WindScanner 2 for Case 2c, generated by calculating the magnitude-squared coherence between the time series, using 100 overlapping Hann windows. Below 1 Hz there is an excellent coherence between hot wire and WindScanner, which means that all coherent structures with scales up to 1 Hz can be measured adequately in agreement by both sensors. Based on this coherence curve, we estimate the coherence-based cut-off frequency of the WindScanner measurement as the frequency at which the coherence drops down to a value of 0.5, indicated with a blue dot in Fig. 20, i.e. $f_{cc} = 2.6$ Hz for this case. When having a closer look at the frequency spectrum in Fig. 19, it can be seen that this



value corresponds quite well to the point at which the WindScanner spectrum begins to deviate from the hot wire spectrum. This also holds true for the spectra of the other cases.

**Table 5.** Estimated probe volume and coherence-based cut-off frequencies, $f_c$ and $f_{cc}$ respectively, of the spectrum of WindScanner 2 with respect to the hot wire.

| Case | 1a | 1b | 2a | 2b | 2c | 2d | 2e |
|---|---|---|---|---|---|---|---|
| Probe volume cut-off frequency $f_c$ [Hz] | 35.6 | 36.4 | 8.5 | 17.1 | 26.3 | 35.0 | 40.5 |
| Coherence-based cut-off frequency $f_{cc}$ [Hz] | 3.3 | 4.7 | 1.0 | 2.1 | 2.6 | 3.6 | 3.7 |
| Ratio $f_c/f_{cc}$ [-] | 10.8 | 7.7 | 8.5 | 8.1 | 10.1 | 9.7 | 10.9 |

In Table 5 the cut-off frequencies of all seven cases are summarised. The probe volume cut-off frequency $f_c$ is estimated according to Taylor's frozen turbulence hypothesis (Taylor, 1938), as explained before in Sect. 4.1. There is a characteristic ratio of 7.7 to 10.9 between the two definitions of the cut-off frequency, which means that Taylor's frozen turbulence hypothesis in combination with the probe volume filtering is not sufficient to explain the lidar measurement principle.

So far we have only analysed the $u$-component of the wind in detail, since it was possible to compare the WindScanner
measurement with the respective one-dimensional hot wire measurement. In Figs. 21 and 22 the dual-Doppler reconstructed $v$-component of the WindScanner measurement is plotted for Case 2c, displaying the entire 10-minute time series and a 20 s snapshot, respectively. In Fig. 21 the repetitive protocol can be recognised, meaning that the generated gusts have a deterministic effect on the $v$-component of the flow. However, zooming in on the time frame between $t = 200$ s and $t = 220$ s in Fig. 22 does not provide a triple repetition of any pattern as it did for the $u$-component in Fig. 15. In conclusion, the gust
protocol is noticeable in the $v$-component, but it is not as consistent as for the $u$-component.

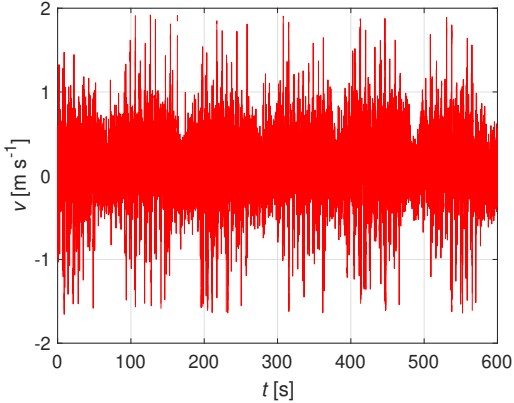

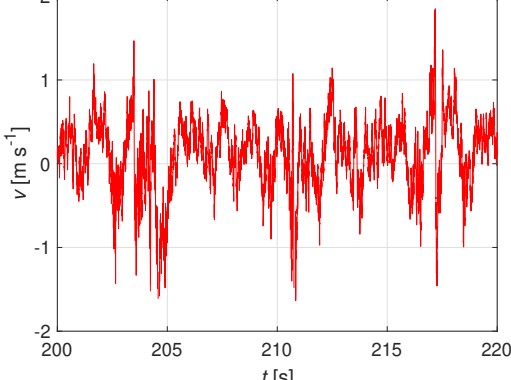

**Figure 21.** The 10-minute time series of the dual-Doppler reconstructed $v$-component for Case 2c.

**Figure 22.** Snapshot of the 10-minute time series of the dual-Doppler reconstructed $v$-component for Case 2c.





**Table 6.** Statistics of the WindScanner dual-Doppler reconstructed $v$-component of the wind.

|  | $\mu_v$ [m s$^{-1}$] | $\sigma_v$ [m s$^{-1}$] | $\mu_v/\mu_u$ [%] | $\sigma_v/\sigma_u$ [%] |
|---|---|---|---|---|
| Case 2a | 0.01 | 0.19 | 0.3 | 53 |
| Case 2b | 0.10 | 0.27 | 2.2 | 40 |
| Case 2c | 0.14 | 0.36 | 1.9 | 38 |
| Case 2d | 0.17 | 0.47 | 1.8 | 39 |
| Case 2e | 0.21 | 0.50 | 1.9 | 37 |

The statistics of the $v$-component for this time series, and for all other cases, are listed in Table 6. For Case 2c the $v$-component has a mean of 0.14 m s$^{-1}$ and a standard deviation of 0.36 m s$^{-1}$. The statistics of the $v$-component are put in perspective by comparison to the $u$-component. By investigating the quotients $\mu_v/\mu_u$ and $\sigma_v/\sigma_u$, it can be concluded that
mean and standard deviation of the $v$-component correspond to approximately 2% and 40% of their respective $u$-component statistics, with the exception of Case 2a, which had the lowest wind speed of $\mu_u = 2.36$ m s$^{-1}$.

The positive mean $v$-component of the flow could be attributed to the asymmetrical setup in the wind tunnel, where more wind tunnel sections were placed near the wall to the right ($y < 0$ m) of Fig. 5 than near the left wall ($y > 0$ m), or to the downstream flow expansion effect.

**4.3   Modelling of WindScanner Characteristics in the Frequency Domain**

After having verified the basic statistics, now the model described in Sect. 3.2 is implemented and evaluated. Figures 23, 24 and 25 correspond to the spectra for WindScanner 2 shown earlier in Figs. 11, 12 and 19, respectively, but are now extended with two additional curves each; the modelled lidar spectrum for a perfect alignment of $\beta = 0°$ excluding noise, shown in green, only taking into account the Lorentzian filter that resembles the probe length averaging, and the modelled lidar spectrum for the
actual case with a misalignment of $\beta = 15.1°$ (see Table 3) including noise, shown in cyan, taking into account the Lorentzian filter as well as a randomly generated white noise term, resembling the full model according to Eq. (10). Please note that the latter curve is not valid for large-scale structures (low frequencies), and is therefore only plotted for the region above 1 Hz.





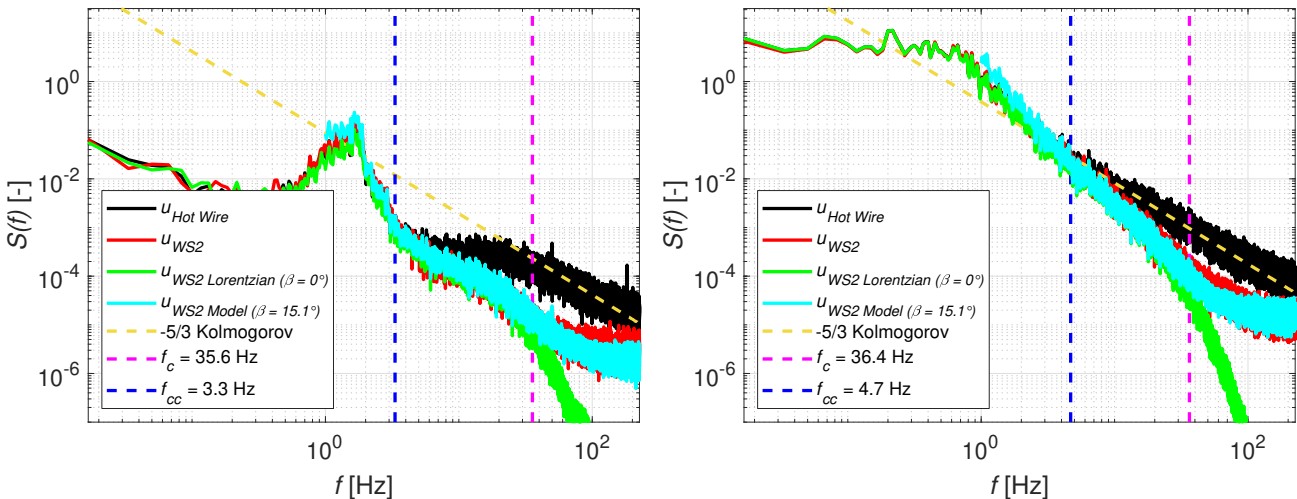

**Figure 23.** Spectrum of WindScanner 2 (measurements and model) for a turbulence intensity of 3% (Case 1a).

**Figure 24.** Spectrum of WindScanner 2 (measurements and model) for a turbulence intensity of 22% (Case 1b).

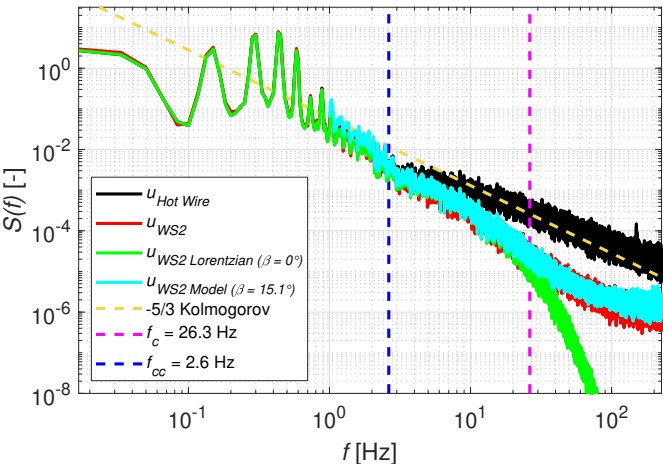

**Figure 25.** Spectrum of WindScanner 2 (measurements and model) for a coefficient of variation of 12.9% (Case 2c).

For all three cases it can be seen that the Lorentzian filter model excluding white noise follows the actual WindScanner
measurement spectrum, until the probe volume cut-off frequency $f_c$ is reached, after which it drops significantly. This means
that the smallest fluctuations, which are within the lidar's probe volume, are filtered out completely. This implies a discrepancy
with the measured spectrum for the highest frequencies, where the lidar measurement does not show such a harsh drop-off.
When extending the model to its complete form, which includes both the 15.1° misalignment and the noise term $\sigma_\eta^2 S_\eta(f)$, we
find a spectrum that closely resembles the measured WindScanner 2 spectrum in a qualitative way.





For the definition of the standard deviation of the white noise, the function of the energy dissipation rate and the mean wind speed (see Eq. (11)) proved to be useful. The performance can be addressed by looking at the difference between the Lorentzian filter model excluding noise and the full model including noise in Figs. 23, 24 and 25 for the frequency region $f > f_c$, which shows a good agreement between the measured and modelled spectrum. However, the model was manually tuned for only seven cases in a controlled environment and for a single lidar configuration with a fixed misalignment angle $\beta$. In order to

consolidate the model, additional data for a wider variety of atmospheric conditions and misalignment angles is required.

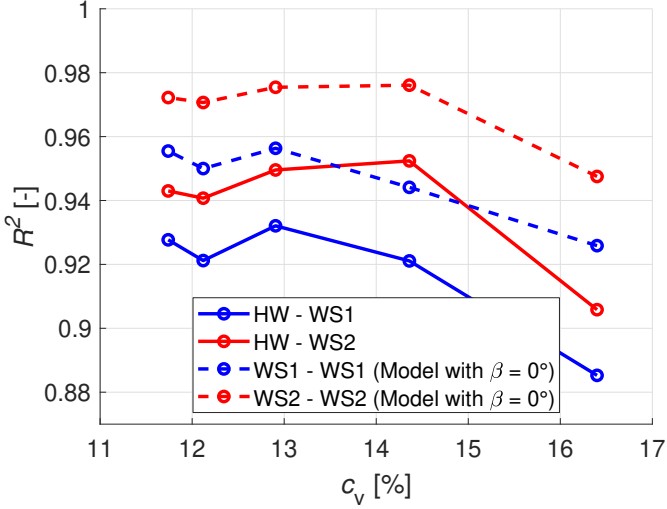

**Figure 26.** Correlation between WindScanner and hot wire time series (solid lines), and between modelled and actual WindScanner time series (dashed lines), as a function of coefficient of variation.

As an additional verification of the approach, we address the correlation between the modelled WindScanner time series with the actual measurement (see Fig. 26). The solid blue and red lines resemble the exact same curves as in Fig. 17 as a reference. The dashed lines correspond to the respective modelled WindScanner time series. Note that these are the results of the time domain implementation of the model (see Eq. (6)) and therefore do not consider the misalignment with the flow. However,

the white noise term is included. Over the entire range of cases, the modelled lidar measurement correlates better to the actual WindScanner measurement than the non-filtered hot wire time series does, even including the white noise. This confirms that the model was implemented in a physically realistic way and can successfully resemble the cw-lidar.

The potential application of the proposed analytical model is that one could derive the atmospheric turbulence spectrum directly from the cw-lidar measurement, by rearranging Eq. (10). In this paper a hot wire was used as a reference, but the

model only depends on it for the initial tuning of the model through constant $C$ and the calculation of the energy dissipation rate $\varepsilon$. If these parameters are derived through alternative methods, one could potentially resolve the full power spectral density from the cw-lidar measurement alone.





## 5    Conclusions

This paper conducted a thorough analysis on the performance of a WindScanner Doppler lidar setup for a variety of turbulent
wind conditions in a wind tunnel.

It was concluded that the correlation between WindScanner and hot wire was much better for a turbulence intensity of 22%
than for a 3% reference, under similar mean wind speeds. However, this was severely influenced by the nature of the turbulence
fluctuations, as a result of the active grid protocol used to generate the high-turbulent case. An identical active grid protocol was
used to address five different cases, with mean wind speeds varying between 2.36 m s$^{-1}$ and 11.25 m s$^{-1}$, and corresponding
coefficients of variation varying between 11.7% and 16.4%. Here it is found that the correlation improves for higher wind
speeds and lower coefficients of variation, which is regarded as a more realistic tendency for atmospheric turbulence.

The difference of the mean longitudinal wind speed with respect to the hot wire was measured to be down to $-1.1$% for the
dual-Doppler reconstructed wind speed, which resembles a lower spread than for the projected wind speeds based on either
of the individual WindScanners. However, the uncertainty on the estimated mean wind speeds might be significant, since a
correction factor of 0.948 was applied to the hot wire measurements after having discovered a faulty hot wire calibration.

The turbulence spectrum of the WindScanner measurement matches very well with the hot wire spectrum at the lowest
frequencies, corresponding to larger eddy structures of the turbulent flow. However, the WindScanner spectrum starts to deviate
from the hot wire spectrum at frequency which is an order of magnitude lower than the probe volume cut-off frequency. For
this reason an alternative way to define the cut-off frequency is proposed, i.e. the frequency at which the coherence drops below
0.5, which corresponds better to the point where a clear deviation of the lidar power spectrum from the hot wire spectrum takes
place.

A model was established that can be used to simulate a lidar measurement time series or power spectrum based on a probe
length filter resembled by a Lorentzian function, acting on the hot wire time series, and randomly generated white noise.
Not only does the resulting model match the actual lidar measurement qualitatively in the frequency domain, its time domain
implementation also correlates better to the actual WindScanner measurement than the non-filtered hot wire time series does.
Modelling of the lidar measurement solely with a Lorentzian probe volume filter was proven insufficient, since also the angle $\beta$
between the line-of-sight pointing direction and the mean flow direction and random measurement noise define the shape of the
measured spectrum. The proposed model, which does cover these aspects, can potentially be used to resolve the power spectral
density function of atmospheric flow on the basis of cw-lidar measurements alone, overcoming the inherent shortcomings of
the measurement principle.

## Appendix A

According to Kristensen et al. (2011) an analytical approach for the spectral density transfer function of a continuous-wave
lidar, measuring a flow within isotropic three-dimensional turbulence within the inertial sub-range under a fixed angle, can be





expressed as such;

$$f(\beta, k, L) = \frac{1}{5\sqrt{\pi}} \frac{\Gamma(1/3)}{\Gamma(5/6)} \sin^{5/3}(\beta)(kL)^{-5/3} + \frac{9}{110}(7\cos(2\beta) - 5)e^{-kL}(kL)^{-2/3}, \qquad (A1)$$

where $\beta$ represents the angle between the lidar's line-of-sight direction and the mean flow direction, $k$ is the wave number, $L$ is the probe length of the lidar measurement and $\Gamma$ is the Gamma function.

In the next step we express the spectral density function for an infinitesimally small probe volume ($L \to 0$) and for a perfectly aligned measurement ($\beta = 0°$):

$$f(0, k, L)\,|_{L\to 0} = \frac{9}{55}(kL)^{-2/3}. \qquad (A2)$$

This expression allows us to define the theoretical spectral transfer function as:

$$T(\beta, k, L) = \frac{f(\beta, k, L)}{f(0, k, L)\,|_{L\to 0}}. \qquad (A3)$$

Substituting for $f(\beta, k, L)$ and $f(0, k, L)\,|_{L\to 0}$ yields:

$$T(\beta, k, L) = \frac{11}{9\sqrt{\pi}} \frac{\Gamma(1/3)}{\Gamma(5/6)} \sin^{5/3}(\beta)(kL)^{-1} + \frac{1}{2}(7\cos(2\beta) - 5)e^{-kL}. \qquad (A4)$$

The last step is the expression of the transfer function as a function of frequency, i.e. $T(f, \beta)$. We make use of the definition of the wave number $k$;

$$k = \frac{2\pi}{\lambda} = \frac{2\pi f}{u_\infty}, \qquad (A5)$$

where $\lambda$ is the wavelength and $u_\infty$ is the free-stream flow velocity.

We can now further simplify the formula by taking into account the previously defined cut-off frequency $f_c$ according to
Eq. (14):

$$f_c = \frac{1}{2}\frac{u_\infty}{L}. \qquad (A6)$$

Combining the aforementioned equations we can define;

$$kL = \pi\frac{f}{f_c}, \qquad (A7)$$

which allows us to express the spectral transfer function as:

$$T(\beta, f) = \frac{11}{9\sqrt{\pi}} \frac{\Gamma(1/3)}{\Gamma(5/6)} \sin^{5/3}(\beta)\left(\pi\frac{f}{f_c}\right)^{-1} + \frac{1}{2}(7\cos(2\beta) - 5)e^{-\pi\frac{f}{f_c}}. \qquad (A8)$$

For the special case in which $\beta = 0°$, corresponding to the situation where the lidar's line-of sight direction is aligned with the mean wind direction, the transfer function reduces to the simple form:

$$T(f) = e^{-\pi\frac{f}{f_c}}. \qquad (A9)$$



*Author contributions.* Marijn Floris van Dooren was in charge of the design and execution of the measurement campaign, the measurement
analysis and paper writing. Anantha Padmanabhan Kidambi Sekar assisted during the measurement campaign, was responsible for installing
the hot wire anemometer, and helped interpreting the results. Lars Neuhaus created the protocols for the active grid and assisted with its
operation. Torben Mikkelsen assisted with the definition of the spectral density function model and provided reviews. Michael Hölling
developed the concept for the active grid and its application. Martin Kühn was actively involved in setting up the measurement concept,
thorough reviewing of the manuscript and had a supervisory role.

*Data availability.* The data used for the analysis in this paper can be made available on request.

*Competing interests.* The authors declare no conflict of interest. The founding sponsors had no role in the design of the study; in the
collection, analyses, or interpretation of data; in the writing of the manuscript, and in the decision to publish the results.

*Acknowledgements.* This work is partly funded by the Federal Ministry for Economic Affairs and Energy according to a resolution by the
German Federal Parliament in the scope of research project DFWind (Ref. Nr. 0325936C). This work is partly done in the scope of the
research project ventus efficiens (Ref. Nr. ZN3024), which is funded by the Ministry for Science and Culture of Lower Saxony through the
funding initiative Niedersächsisches Vorab. Furthermore we would like to thank our colleagues Stephan Voß, Paul Hulsman and Frederik
Berger for their assistance during the extensive WindScanner measurement campaign. Besides that we would like to express our gratitude to
David Bastine for the discussions on the spectral analysis. Lastly we would like to thank Mikael Sjöholm and Claus Brian Munk Pedersen
from the Technical University of Denmark for their invaluable remote assistance with troubleshooting WindScanner hardware- and software-
related challenges during the measurement campaign.





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
