# Peer review of "Modelling the Spectral Shape of Continuous-Wave Lidar Measurements in a Turbulent Wind Tunnel"

_Atmospheric Measurement Techniques, 2021_

## Referee Comment (RC1)

**Review of the manuscript AMT-2021-233, entitled "Modelling the Spectral Shape of Continuous-Wave Lidar Measurements in a Turbulent Wind Tunnel", by van Dooren M. F., Sekar A. P. K., Neuhaus L., Mikkelsen T., Hölling M., Kūhn M.**

**General comments**

The study of van Dooren *et al*. addresses the problem of the spatial averaging effect performed by a continuous-wave lidar on the measured turbulent flow. This effect is quantified through a novel experimental technique, i.e. using a short-range Lidar in a wind tunnel environment and assessing it against hot-wire anemometry. The analysis is well performed by taking into account the most important aspects of the problem. Writing sometimes is unclear and colloquial; thus, some comments are reported in the following to improve clarity.

**Specific comments**

L94: Provide some details about the seeding procedure (e.g., seed type, mean particle mass and volume, point(s) of application, any measurement about the concentration etc.).

L125: Provide a quantification of the time lag and, if possible, a plot reporting the mentioned cross-correlation.

L145: The equality between the full-width half-maximum length (FWHM) and probe length holds for a continuous wave-lidar, whereas for a pulsed lidar those are distinct parameters. Please specify them.

L163: It is more correct to state that their contributions, weighted by the respective sine and cosine functions, are negligible with respect to $u_p$.

L212-213: Do you have any reference assessing this assumption?
In any case, this procedure does not make much sense to me as, in case of lidar measurements, the noise is related to random fluctuations of the backscattered signal, which introduces an uncertainty in the Doppler shift (see e.g. Frehlich and Yadlowsky, 1994; Frehlich, 1997 for pulsed Doppler lidar). Hypothesizing a connection between the noise and physical properties of the turbulent flow is a strong statement that needs to be better justified. As validation, you can quantify the noise from the lidar data in an independent way, for instance the auto-correlation method of Lenshow *et al*. (2000).

L253: Showing a correlation between instantaneous values may be questionable as the latter are affected by the uncertainty due to the lidar noise (which has not been removed) and the interpolation of the hot-wire signal onto the lidar time stamp. If you want to compare the recorded time distributions, I recommend to at least perform a moving average of the signals and then apply the linear regression. As an alternative, you can calculate mean velocity and standard deviation over non-overlapping periods (whose length must be carefully established) and then compare them.

L319: As for Figures 9 and 10, I recommend to compare either a moving average over a short time period.

**Technical corrections**

L2: Add the meaning of the acronym "lidar".

L4-6: Please consider to change the statement to: "The hot-wire anemometer is used as theoretical reference to assess the lidar-based statistics, time series and spectra". Remove the mention to the Taylor hypothesis as the spectra are evaluated in frequency.

L22: Please add some references to important wind tunnel studies of wind turbines.

L32: Please state that, in contrast to the probing techniques mentioned in the previous paragraph, the lidar technology has been originally developed for real-scale studies and you are proposing a novel implementation of this technology.

L34: Replace "[…] but make up for it […]" with "[…] but, on the other hand, […]".

L55: Please add a short paragraph to describe the content of the next Sections.

L71-74: Replace "However, for the measurement campaign described in this paper, they are placed near the walls of the wind tunnel. Three of them can be seen on the right side of the nozzle in Fig. 1. The two remaining ones are parked at the back of the wind tunnel and serve as measurement platforms for the lidars, as illustrated by Fig. 2" with: "For the present campaign, only two test sections are used as measurement platforms for the lidars, as illustrated by Fig. 2."

L78: Specify that two identical continuous-wave lidars are used in this campaign.

L81: Specify that the Doppler shift is calculated with respect to the emitted laser frequency.

L135: Add reference to Sjöholm *et al*. (2009).

L187: I think here you are referring to Eq. (7). If so, please correct.

L219: Please add: "The Kolmogorov spectrum in the inertial subrange is modeled as follows: […]".

L226: Specify that, at this stage, the comparison is done in time between instantaneous values.

L243: Please add that this difference will be addressed in the following part of the Subsection.

L350: It is incorrect to state that the low-frequency peaks do not have physical significance. I would change this sentence with "As they result from external gust variations, these peaks are not deemed to be due to turbulence […]".

L358: For the sake of clarity, please report the definition of coherence here.

L386: To my understanding, here you are applying the Lorentzian model described in Sec. 3.2 to the hot-wire spectrum and qualitatively compare the similarity with the WindScanner 2 streamwise spectrum. Please state this clearly at the beginning of the Subsection.

**References:**

- Frehlich, R. G., and M. J. Yadlowsky. "Performance of mean-frequency estimators for Doppler radar and lidar." *Journal of atmospheric and oceanic technology* 11.5 (1994): 1217-1230.

- Frehlich, R. G. "Effects of wind turbulence on coherent Doppler lidar performance." *Journal of Atmospheric and Oceanic Technology* 14.1 (1997): 54-75.

- Lenschow, D. H., Volker W., and Christoph S.. "Measuring second-through fourth-order moments in noisy data." *Journal of Atmospheric and Oceanic technology* 17.10 (2000): 1330-1347.

- Sjöholm, M., Mikkelsen, T., Mann, J., Enevoldsen, K., & Courtney, M. (2009). Spatial averaging-effects on turbulence measured by a continuous-wave coherent lidar. *Meteorologische Zeitschrift*, *18*(3), 281–287.

---

## Referee Comment (RC2)

**General Comments:**

The article is a strong contribution on a novel topic that deserves publication in AMT. The authors first verify that a dual-doppler lidar situated in a wind tunnel can recover adequately the mean and spectral character of hotwire data, given certain limitations due to the pointing direction and volume averaging of the lidar. The discovery of a 0.5 coherence level as the frequency threshold of valid lidar spectra rather than a threshold based on Taylor's hypothesis appears novel. They then describe the validation of a new model for the lidar-derived spectrum. The spectral model with misalignment and noise is clearly an improvement over the plain Lorentzian model without misalignment (c.f., Figures 23-25), and offers a new path to get an unfiltered u spectrum from a Doppler lidar measurement. Despite the presence of an admittedly large calibration error that casts doubt on some results, the overall objective of the article is still met.

I do have three main concerns in addition to the line by line comments, which elaborate further:

1. The repeated reference to better correlation for TI=22% than TI=3%...I believe the RMSE is lower for the TI=3%, and the better R^2 correlation of the high TI case is just a function of the range of velocities used in the linear regression. The authors' explanation about relatively higher energy content in the low frequency range (that the lidar can resolve well) for the higher TI case is understood, but the reviewer wonders if this is really a critical piece of the puzzle or not. See my comments further below.
2. The discussion and reasoning around equation 11 is a little bit sparse and not completely intuitive.
3. Dual doppler vs single doppler. It is not entirely clear to me that the dual doppler data belongs in this paper, since the main results in this article only use WS2 (because the spectral transfer function is for a single doppler only). I would ask that the focus on the dual doppler reconstruction be removed or justified better. Moreover, the authors state in more than one place that the dual-doppler reconstruction is better than either of the individual wind scanners for the u component of velocity. I don't think this is supported by Figure 17/18 or Table 4. If the dual doppler is NOT better, this doesn't overshadow the usefulness of a dual-doppler technique which can resolve two components of velocity, but the text appears misleading about the u component results. (I also noticed that the dual-doppler results are not even mentioned in the abstract, which further makes me wonder: why even have the dual doppler results in this paper?)

**Specific Comments:**
*Abstract:*

- No comments.

*Introduction:*

- Well written.

*Methodology, Part I*

- Well written.

*Methodology, Part II*

- Lines 173-174: does the work of Sjoholm et al. and Angelou et al. specifically describe the higher frequencies in the lidar spectra as white noise? While it may seem obvious to some, there is not strong justification given for why white noise (i.e., the Gaussian distribution) is chosen. As this is a main advance of the paper, I think a little more information is warranted.
- Line 212: it is not self-evident that sigma_n should be a function of energy dissipation rate or wind speed. Is the energy dissipation and wind speed related to the decorrelation time? Is this measurement shot-noise limited? Could you explain this more?
- Equation 11: Why is gravity a relevant variable in the dimensional analysis?

*Results and Discussion*

Comparisons between WindScanner and hotwire…

- Line 255: I don't know if I would put so much weight on R^2 values here. I wonder how the RMSE compares between figures 9 and 10. The larger spread of u from a turbulent field seems to be giving higher R^2 even though there is clearly more absolute variation between u_p and u over most of the range for the more turbulent case. If you were to run the lower TI case at a freestream velocity of both 5 m/s and 15 m/s (i.e., over the same range as shown for the high TI case), the R^2 of the combined data for the low TI case would be larger than for the high TI case, right?
- Line 284-285: I understand that you are saying the small scales play a more dominant role for Figure 11 than Figure 12, which seems true based on the low frequency amplitudes of S(f). The energy content at f_cc is still more than 10 times lower than at lower frequencies for Fig. 11, though. Why don't you integrate Figures 11 and 12 from 0 to f_cc and from f_cc to infinity. See what fraction of the turbulence is not fully resolvable by the lidar and report this rather than emphasizing the difference in R^2 values, which doesn't seem as relevant to me.
- Line 289: I was expecting this line to say, "A possible reason is the insufficiency of the full-width half maximum metric to characterize the effective probe length." Don't you agree? What about the implicit assumption that the turbulence is isotropic, could this also be a possible culprit?

Analysis of WindScanner measurements of turbulent gusts

- Line 325: this is the first time you've mentioned 1 Hz averaged time series. Could you please give a brief mention of why you perform this time averaging (I assume to get out of the small eddy range that can't be resolved by the lidar).
- Line 331: Can you comment on why the green line is not the highest for the 1 Hz data. Is it that at 452 Hz, the two lidars are both filtering small-scale turbulence and thus agree quite closely compared to the unfiltered HW, but at 1 Hz, both the hotwire and lidar are on more even playing field and can both resolve all the scales?
- Line 338: The lowest errors appear to be found for the blue line not the black line, and a quick subtraction of the columns in Table 4 suggests that the mean difference between WS1 and HW is smaller than between WS and HW. Please revise this statement or justify it. Is the mean error of the dual-doppler reconstruction related to the fact that the hot-wire only measures one component? Why is validating the dual-doppler reconstruction given so much weight in this paper?

- Line 342: you say there is a "bias between WindScanner 1 and 2, which increases linearly with the mean wind speed". This is not obvious from the plot except moving from \mu ~2 to ~5 where the gap widens between red and blue. Please revise or justify.
- Line 350: you say, "they do not have physical significance". Please clarify your statement about physical significance as this is clearly a physical phenomenon in the flow that is being resolved by both measurement systems.
- Table 5: I wonder if the ratios of f_c/f_cc in this table are possibly more important in the long run than the 0.5 coherence observation, since in a real application of this technique, you will not have a reference instrument to calculate coherence, right? Would it be appropriate to suggest that the effective probe volume given by the FWHM maximum could be at most an order of magnitude in error based on this data?
- Line 375: I think this is a good conclusion to draw. It looks like the amplitude of the protocol-induced gust is larger in the u rather than v direction – could this be a reason why the triple repetition is being lost in Figure 22? Reading further, I see that these differences are quantified in Table 6. If you believe my argument, I think you could comment on how the fact that sigma_v/sigma_u << 100% might be related to your conclusion in line 375.

Modeling of WindScanner characteristics in the frequency domain

- Line 392: you say, "the latter curve is not valid for large-scale structures". Just to clarify, is this because it is only derived for the inertial subrange?
- Line 415: the potential application is very interesting and seems worthwhile. You have used the word "atmospheric" twice in the last three paragraphs. From the introduction of the paper, I was under the impression that you want to use the dual lidar technology in wind tunnel studies of wind turbine configurations? Could you clarify here (and in the abstract/introduction) if your aim is for wind tunnel or field measurements (or both)?

*Conclusion*

- Line 427: In reference to 1.1%, you say that the dual doppler gives "lower spread". However, the 1.1% comes from an analysis of mean error, not scatter. Could you clarify this wording?

**Technical Comments**
- Figure 1 might be more useful if it included a zoom in of the nozzle with the active grid.
- Line 68: add "to" before reproduce
- Table 2: I think it would be appropriate to give the names of the variables in Table 2 and not just the symbols.
- Line 140: you mention that L is the probe length twice.
- Figure 11/12: Could you note in the caption that f_cc will be defined later in Section X…?
- Lines 327-328: no need to describe what the different colored lines mean since it's in the figure.
- Lines 346: no need to describe the line colors in the text.
- Line 427: "down to -1.1%." Can you say "within 1.1%" instead to be more precise?

---

## Author Comment (AC1)

**Author's Response**

On the community comment on "Modelling the Spectral Shape of Continuous-Wave Lidar Measurements in a Turbulent Wind Tunnel" by Marijn Floris van Dooren et al., Atmos. Meas. Tech. Discuss., https://doi.org/10.5194/amt-2021-233-CC1, 2021

26.11.2021

Marijn Floris van Dooren et al.

Dear Etienne Cheynet,

Thank you for taking the time to post a community comment on the pre-print manuscript amt-2021-233 on 17.09.2021. We are very pleased that you regard this as a valuable study for engineers and scientists working on turbulence flow measurement techniques. In this author's response, we will rephrase your remarks and questions in blue and answer them in black.

1. The studies present some coherence measurements, which I found really interesting, given that similar studies were conducted with the short-range WindScanner system in an outdoor environment in 2014 [1]. The coherence can be defined for longitudinal, lateral and vertical separations. Therefore, it was unclear to me if the studies discussed lateral or longitudinal coherence. Maybe this can be explained in a few lines?

   In this study the coherence that is presented is not representing any specific spatial coherence as a property of the flow but is used as a tool to compare two different sensors with one another. Indeed, there was a 7 cm separation between the WindScanner measurement point and the reference hot wire anemometer, however, the different measurement principles are assumed to have a much higher impact on the coherence graph than this spatial separation in the lateral direction. We included a better explanation in the paper about how this coherence function should be interpreted (**L384-389** in the revised manuscript).

2. The manuscript suggests that the frequency at which the lidar power spectrum deviates from the hot wire reference spectrum is the frequency at which the coherence drops under 0.5. Maybe a more accurate unit than the frequency is the wavenumber. Otherwise, the frequency at which the coherence becomes lower than 0.5 may depend on the mean wind speed. To go even further, the wavenumber multiplied by the separation distance $D$ could be used as the coherence is a function of $D$. Therefore, at large distances, the frequency at which the coherence is under 0.5 will be much lower than at small distances.

   You are right about the relevance of the wave domain when we are speaking about lidar probe volume averaging. In this study we opted for working in the frequency domain, and we assumed Taylor's Hypothesis to hold true, such that there is indeed a direct dependency on the mean wind speed, see Eq. (1):

   $$k = \frac{2\pi}{\lambda} = \frac{2\pi f}{u_\infty}$$

   To be consistent with the spectral analysis in the paper, also performed in the frequency domain, we would like to stick to the current variable.

We acknowledge that the so-defined 'coherence cut-off frequency' will indeed decrease for increasing separation distances. However, as explained in the answer to question 1, we are dealing with only one spatially separated measurement comparing a single WindScanner to the reference hot wire anemometer, and thus the separation distance is not varied in this study.

3.  How does the spectral correction improve the coherence estimates? In [1], it was suggested that since the coherence is a normalized spectral characteristic, the spatial averaging effect has a limited influence on the coherence estimates. However, in [5], it was also suggested that the spatial averaging may not be negligible if the probe volume is significantly larger than a typical length scale of turbulence.

    Unfortunately, we cannot make a statement on whether a spectral correction on the lidar measurement will improve the spatial coherence estimates, based on this data set, since the definition of coherence we present is the coherence between the WindScanner and the hot wire anemometer.

4.  For engineering applications, one fundamental turbulence characteristic in wind tunnel tests is the integral length scale, which can be calculated with the autocorrelation function. Have you attempted to estimate it with the lidar system? If yes, how does it compare with the hot wire anemometer measurements? In [1], an overestimation by the lidar system was observed. I am curious to know if it is also the case in your study.

    We have not calculated the autocorrelation function based on either the lidar or the reference hot wire anemometer measurement before, however we did investigate it now. We used the approach from [1] to define the integral time scale based on the autocorrelation as such:

$$T = \int_{t=0}^{t(R(t)=0)} R(t)dt$$

Where $R(t)$ is the normalized autocorrelation function of a time series (lidar or hot wire) and $T$ is the integral time scale. Using Taylor's Hypothesis, the integral length scale $L$ follows from the multiplication with the mean wind speed $\bar{u}$:

$$L = \bar{u}T$$

In Figure 1, plots of the autocorrelation function are shown for both lidar and hot wire time series, for cases 1a, 1b and 2c, for the region between $t = 0$ and $t(R(t) = 0)$. Table 1 provides an overview of the calculated integral time and length scales for the three cases.

[Figure]

**Figure 1**: *Plots of the autocorrelation function of lidar and hot wire time series for three cases (1a, 1b and 2c).*

**Table 1**. *Overview of calculated integral time scale and integral length scale for both lidar and hot wire anemometer for three cases (1a, 1b and 2c).*

| Case | $T_{lidar}$ [s] | $T_{hw}$ [s] | $L_{lidar}$ [m] | $L_{hw}$ [m] |
|------|------|------|------|------|
| **1a** | 0.10 | 0.09 | 0.99 | 0.87 |
| **1b** | 0.40 | 0.34 | 3.35 | 4.08 |
| **2c** | 0.45 | 0.43 | 3.22 | 3.12 |

The values for the integral length scale are in the order of meters, much smaller than the values found in [1], which can be attributed to the differences between the flow in the free field and inside the wind tunnel. On average there is no consistent over- or underprediction by the lidar, as it depends on the case.

These results have not been included in the revised paper.

5. Although the purpose of the paper is on the high-frequency correction of the lidar velocity spectrum, the measurement technique presented in the manuscript has a wide range of potential applications in a wind tunnel facility. One of them is the study of wake behind bluff bodies. The short-range WindScanner has been successfully used in the past to study the turbulent flow around bridge decks [2], a tree [3] or a fence [4] in "full-scale". What about scaled models in a controlled environment? Do you think including such a discussion in the manuscript may be relevant to highlight the possible applications of the short-range WindScanner system in wind tunnels in the field of wind engineering, wind energy or fluid mechanic?

We acknowledge the potential applications of short-range WindScanner technology that you mention. Indeed, there already have been measurements of model wind turbine wakes inside a wind tunnel [6, 7] that successfully demonstrated this application. These references and more are mentioned in the introduction (**L39-43** in the revised manuscript). However, the main objective of the paper is the modelling of the lidar's measured spectrum in case of undisturbed flow, without any objects of study placed inside the wind tunnel. Therefore, we would like to keep the discussion focussed on the further verification of the model for different wind conditions in both the wind tunnel and in the open field.

**References**

[1] Cheynet, E., Jakobsen, J.B., Snæbjörnsson, J. et al.: Application of short-range dual-Doppler lidars to evaluate the coherence of turbulence, Exp. Fluids, 57, 184, https://doi.org/10.1007/s00348-016-2275-9, 2016.

[2] Cheynet, E., Jakobsen, J. B., Snæbjörnsson, J., Angelou, N., Mikkelsen, T., Sjöholm, M., & Svardal, B.: Full-scale observation of the flow downstream of a suspension bridge deck. Journal of Wind Engineering and Industrial Aerodynamics, 171, 261-272, 2017.

[3] Angelou, N., Mann, J., & Dellwik, E.: Scanning Doppler lidar measurements of drag force on a solitary tree. Journal of Fluid Mechanics, 917, 2021.

[4] Peña, A., Bechmann, A., Conti, D., & Angelou, N.: The fence experiment – full-scale lidar-based shelter observations, Wind Energy Science, 1(2), 101-114, 2016.

[5] Debnath, M., Brugger, P., Simley, E., Doubrawa, P., Hamilton, N., Scholbrock, et al.: Longitudinal coherence and short-term wind speed prediction based on a nacelle-mounted Doppler lidar. In Journal of Physics: Conference Series Vol. 1618, No. 3, 032051, IOP Publishing, 2020.

[6] van Dooren, M. F., Campagnolo, F., Sjöholm, M., Angelou, N., Mikkelsen, T., and Kühn, M.: Demonstration and uncertainty analysis of synchronised scanning lidar measurements of 2-D velocity fields in a boundary-layer wind tunnel, Wind Energ. Sci., 2, 329–341, https://doi.org/10.5194/wes-2-329-2017, 2017.

[7] Hulsman, P., Wosnik, M., Petrović, V., Hölling, M., and Kühn, M.: Development of a Curled Wake of a Yawed Wind Turbine under Turbulent and Sheared Inflow, Wind Energ. Sci. Discuss. [preprint], https://doi.org/10.5194/wes-2021-65, in review, 2021.

---

## Author Comment (AC2)

**Author's Response**

On the first referee comment on "Modelling the Spectral Shape of Continuous-Wave Lidar Measurements in a Turbulent Wind Tunnel" by Marijn Floris van Dooren et al., Atmos. Meas. Tech. Discuss., https://doi.org/10.5194/amt-2021-233-RC1, 2021

26.11.2021

Marijn Floris van Dooren et al.

Dear Sir/Madam,

Thank you very much for taking the time to review our pre-print manuscript and for your helpful feedback and questions. We will rephrase your comments in blue and include our response in black.

**Specific comments**

**L94**: Provide some details about the seeding procedure (e.g., seed type, mean particle mass and volume, point(s) of application, any measurement about the concentration etc.).

We added the following description of the seeding procedure to the paper: 'In order to maintain a sufficient number of aerosols in the wind tunnel to reflect the laser beam, seeding with Di-Ethyl-Hexyl-Sebacat (DEHS) was applied every few hours with a PALAS AGF 10.0 liquid nebulizer at the back of the wind tunnel, using the closed return wind tunnel itself for circulation. DEHS has a density of 0.91 g/cm$^3$ and a mean particle diameter of 0.5 µm. The aerosol concentration was not confirmed by measurement; however, the quality of the WindScanners' backscatter signal was used as an indirect indicator'.

**L125**: Provide a quantification of the time lag and, if possible, a plot reporting the mentioned cross-correlation.

We have quantified the time lag for the three cases shown in the paper (1a, 1b and 2c) and provided plots of the cross-correlation. The time lag includes the deviation in the clock time between the respective WindScanner and hot wire computers, but more importantly, illustrates the delay in activation of the hot wire logger after the WindScanners. We consequently started the WindScanner measurement first, and afterwards activated the hot wire measurement data logger for a duration of exactly 10 minutes. This procedure guaranteed that we consistently had full 10-minute data sets for both anemometers.

[Figure]

**Figure 1**: *Plots of the cross-correlation function between lidar and hot wire time series for three cases (1a, 1b and 2c), used for the temporal synchronisation.*

Please note that in all cases the time lag can be confirmed convincingly. Especially for the high turbulence case (1b) the cross-correlation function is unambiguous. For case 2c there are multiple high peaks because of the repetitive gust protocol, but the peak closest to $\Delta t = 0$ s is the relevant one.

Although the temporal synchronisation between the WindScanners and the hot wire are an important aspect of the data analysis, we regard this as a trivial procedure. Therefore, we would like to refrain from adding these plots to the paper itself.

**L145**: The equality between the Full Width at Half Maximum (FWHM) and probe length holds for a continuous wave-lidar, whereas for a pulsed lidar those are distinct parameters. Please specify them.

We are aware that the definitions for probe length and Full Width at Half Maximum length are different between cw-lidar and pulsed lidar technology. In order to avoid confusion, we would like to keep only the definitions for cw-lidar in the paper. However, it is explicitly mentioned, where applicable, that the definition is only valid for cw-lidar (e.g. **L143** and **L150** in the revised manuscript).

**L163**: It is more correct to state that their contributions, weighted by the respective sine and cosine functions, are negligible with respect to $u_p$.

Thank you for pointing out that it is more reasonable to assume that $\sin \chi \cos \delta \, v$ and $\sin \delta \, w$ are negligible with respect to $u_p$ rather than assuming $v = 0$ and $w = 0$. We agree with this and have corrected this statement in the paper accordingly.

**L212-213**: Do you have any reference assessing this assumption? In any case, this procedure does not make much sense to me as, in case of lidar measurements, the noise is related to random fluctuations of the backscattered signal, which introduces an uncertainty in the Doppler shift (see e.g. Frehlich and Yadlowsky, 1994; Frehlich, 1997 for pulsed Doppler lidar). Hypothesizing a connection between the noise and physical properties of the turbulent flow is a strong statement that needs to be better justified. As validation, you can quantify the noise from the lidar data in an independent way, for instance the auto-correlation method of Lenshow *et al*. (2000).

We were not able to identify references that suggest a relationship between the lidar spectral noise level and flow parameters such as the energy dissipation rate and mean wind speed. However, we found very clear indications that such a connection does exist and would like to elaborate on it. First, we started with the assumption that the noise in a lidar measurement should be related to random fluctuations of the backscattered signal only, and that this is a property inherent to the lidar measurement principle and not to the physical properties of the turbulent flow. However, in our analysis we saw a convincing increase of the noise level for more energetic flows with higher wind speeds. We have evaluated various possible dependencies. The following lists the steps describing our empirical analysis of the lidar spectral noise estimate:

1.  For each case (1a, 1b and 2a-2e) we manually tuned the lidar noise standard deviation $\sigma_\eta$ to the model in Eq. (10) for the best possible match between modelled and measured lidar spectrum.
2.  With a linear regression, we then tried to identify a parameter or a combination of parameters that could best match those tuned values for $\sigma_\eta$.
3.  In the end the best fit was found for the square root of the product of energy dissipation rate $\varepsilon$ and mean wind speed $\mu_u$.
4.  We tried to make the units match by including physical constants, of which the gravitational acceleration parameter seemed to be the best candidate. However, the second referee was

sceptical about the inclusion of this unrelated parameter, which made us decide to leave it out and accept a constant with a unit instead, indicating that there might still be unidentified parameters playing a role in the estimation of the lidar spectral noise level.

Figure 2 shows the relationship of $\sigma_\eta$ with the mean wind speed $\mu_u$ and the standard deviation $\sigma_u$. The fit is not convincing, although the dimensions match.

[Figure]

**Figure 2**: *Plots of the relationship of the lidar spectral noise standard deviation $\sigma_\eta$ with the mean wind speed $\mu_u$ and the standard deviation $\sigma_u$, respectively.*

Figure 3 displays the relationship of $\sigma_\eta$ with three different definitions of the coefficient of variance $c_v$, which is like the standard deviation $\sigma_u$, but only considers the small-scale fluctuations, which are most likely to influence the lidar noise. The three plots look similar, although the absolute values are different. The fit is a significant improvement compared to the standard deviation $\sigma_u$ of the full time series. The unit matches to m/s.

[Figure]

**Figure 3**: *Plots of the relationship of the lidar spectral noise standard deviation $\sigma_\eta$ with the coefficient of variation $c_v$ calculated in different ways; Left: Difference between the modelled lidar time series (only Lorentzian filter without added noise) and the hot-wire time series. Middle: Integrated coefficient of variance of the hot-wire spectrum from $f_c$ to infinity. Right: Integrated coefficient of variance of the hot-wire spectrum from $f_{cc}$ to infinity.*

Finally, Fig. 4 shows the more convincing relationships that include the energy dissipation rate $\varepsilon$ (with the unit m²/s³), both alone and in combination with the mean wind speed $\mu_u$.

[Figure]

**Figure 4**: *Plots of the relationship of the lidar spectral noise standard deviation $\sigma_\eta$ with the square root of the energy dissipation rate and the product of it with the mean wind speed, respectively.*

Based on this thorough empirical analysis, we decided that there is a convincing relationship between the noise standard deviation and the energy dissipation rate in combination with the mean wind speed of the flow. Our interpretation of this phenomenon is that more energetic flows, with a higher energy dissipation rate, inhabit more pronounced fluctuations and gradients in the probe volume of the cw-lidar, which result in a higher uncertainty in the estimated wind speed that influences the small-scale fluctuations in the cw-lidar time series, regardless of the Lorentzian low-pass filter effect.

This hypothesis needs more elaboration. Most likely the white noise is partly due to shot noise of the lidar, and partly related to global flow parameters. Please note that we are not suggesting the white noise is in any way correlated to the flow in the time domain, it is mainly the notion that global flow parameters could influence the absolute value of the lidar noise standard deviation. The model is not yet complete and might rely on further flow parameters, e.g. time and length scales, and lidar parameters.

Since we are convinced that this could be an interesting finding, we would like to keep the adjusted model described by Eq. (11) in the paper. Currently the empirical analysis is not included in the paper itself, but if this is deemed necessary, we will add (part of) this elaboration as an appendix.

**L253**: Showing a correlation between instantaneous values may be questionable as the latter are affected by the uncertainty due to the lidar noise (which has not been removed) and the interpolation of the hot-wire signal onto the lidar time stamp. If you want to compare the recorded time distributions, I recommend to at least perform a moving average of the signals and then apply the linear regression. As an alternative, you can calculate mean velocity and standard deviation over non-overlapping periods (whose length must be carefully established) and then compare them.

Instead of correlating instantaneous (and linearly interpolated) measurement values, we agree with your suggestion and performed a moving average before creating the correlation plots. We used a window of 20 samples, since this way the effective averaging of the 451.7 Hz time series will yield a smoothed time series where frequencies above ~22.6 Hz are filtered out. This value is just below the lowest cut-off frequency of the lidar measurement modelled by means of the Lorentzian filter full width half maximum among the three presented cases. The effect of the smoothing on the goodness of fit coefficient for the three cases portrayed in the paper is listed in Table 1.

**Table 1**: *Improvement of the goodness of fit coefficient for the correlation between the smoothed WindScanner 2 and hot wire time series for three cases (1a, 1b and 2c).*

| Case | Figure | $R^2$ (instantaneous) | $R^2$ (smoothed) |
|------|--------|-----------------------|------------------|
| 1a   | 9      | 0.656                 | 0.790            |
| 1b   | 10     | 0.931                 | 0.957            |

| 2c | 16 | 0.950 | 0.975 |
|---|---|---|---|

The moving average procedure mostly benefits the goodness of fit coefficient for case 1a, while also slightly increasing the already high values of cases 1b and 2c.

**L319**: As for Figures 9 and 10, I recommend to compare either a moving average over a short time period.

See the answer above for **L253**.

**Technical corrections**

**L2**: Add the meaning of the acronym "lidar".

We included the meaning of the abbreviation 'lidar' (light detection and ranging) for the first occurrence in the paper.

**L4-6**: Please consider to change the statement to: "The hot-wire anemometer is used as theoretical reference to assess the lidar-based statistics, time series and spectra". Remove the mention to the Taylor hypothesis as the spectra are evaluated in frequency.

We changed the statement to your proposed alternative. We would still like to mention the 'theoretical spectrum using Taylor's Hypothesis', since the latter is a vital assumption to convert the spectral model, which uses the spatial Lorentzian filter as a basis, from the wave domain to the frequency domain.

**L22**: Please add some references to important wind tunnel studies of wind turbines.

We included three additional references related to wind turbine tests in wind tunnels (Campagnolo *et al*., 2016; Tian *et al*., 2018; Berger *et al*., 2021). Another relevant paper (Bottasso *et al*., 2014) was already mentioned in the same paragraph.

**L32**: Please state that, in contrast to the probing techniques mentioned in the previous paragraph, the lidar technology has been originally developed for real-scale studies and you are proposing a novel implementation of this technology.

We included the sentence 'In contrast to the other aforementioned sensors, lidar technology was originally developed for real-scale studies.', right before lidar measurements in wind tunnels are mentioned (**L35**). We also changed the word 'new' to 'novel' (**L36**).

**L34**: Replace "[…] but make up for it […]" with "[…] but, on the other hand, […]".

We changed the sentence to 'Lidars measure… …aforementioned sensors, but are, on the other hand, a… …measurement technique'.

**L55**: Please add a short paragraph to describe the content of the next Sections.

The structure of the paper is now announced at the end of the introduction.

**L71-74**: Replace "However, for the measurement campaign described in this paper, they are placed near the walls of the wind tunnel. Three of them can be seen on the right side of the nozzle in Fig. 1. The two remaining ones are parked at the back of the wind tunnel and serve as measurement platforms for the lidars, as illustrated by Fig. 2" with: "For the present campaign, only two test sections are used as measurement platforms for the lidars, as illustrated by Fig. 2."

We rewrote this sentence, in a slightly different way than how you proposed, to 'For the present campaign, all test sections are placed near the walls of the wind tunnel, and two of them are used as measurement platforms for the lidars…'.

**L78**: Specify that two identical continuous-wave lidars are used in this campaign.

The first sentence of this paragraph now states that both lidars are identical.

**L81**: Specify that the Doppler shift is calculated with respect to the emitted laser frequency.

We added the following sentence: '…Doppler shift in Hertz. The latter is defined as the difference between the backscattered and emitted laser frequency'.

**L135**: Add reference to Sjöholm *et al*. (2009).

The reference to Sjöholm *et al*. (2009), which was already part of the bibliography, is now included in this line.

**L187**: I think here you are referring to Eq. (7). If so, please correct.

We were indeed referring to Eq. (7) and not Eq. (6), so we corrected this reference accordingly.

**L219**: Please add: "The Kolmogorov spectrum in the inertial subrange is modelled as follows: […]".

We now properly introduced Eq. (12) and relocated the reference to the last sentence: 'The Komolgorov spectrum in the inertial sub-range is modelled by Eq. (12): […]'.

**L226**: Specify that, at this stage, the comparison is done in time between instantaneous values.

It is now stated that the comparison applies to instantaneous values, as such: '…was carried out for instantaneously sampled time series on a 10-minute basis…'.

**L243**: Please add that this difference will be addressed in the following part of the Subsection.

We added the sentence 'This difference will be addressed in the following part of the Subsection'.

**L350**: It is incorrect to state that the low-frequency peaks do not have physical significance. I would change this sentence with "As they result from external gust variations, these peaks are not deemed to be due to turbulence […]".

We agree that it is incorrect to label the frequency peaks as 'no physical significance'. The second referee also pointed this out. We meant that they are not associated with atmospheric turbulence, but that does not mean it is not a physical phenomenon. We gladly accept to use your alternative wording.

**L358**: For the sake of clarity, please report the definition of coherence here.

A precise definition of the coherence, including an equation, is added to the paragraph.

**L386**: To my understanding, here you are applying the Lorentzian model described in Sect. 3.2 to the hot-wire spectrum and qualitatively compare the similarity with the WindScanner 2 streamwise spectrum. Please state this clearly at the beginning of the Subsection.

Your understanding of the implementation of the model and its evaluation is correct. We have added the sentence 'The modelled lidar spectrum is generated by applying the methodology on the measured hot-wire spectrum.' (**L402-403** of the revised paper) to emphasise it more clearly.

**References**

Berger, F., Onnen, D., Schepers, J. G., and Kühn, M.: Experimental Analysis of Radially Resolved Dynamic Inflow Effects due to Pitch Steps, Wind Energ. Sci. Discuss., https://doi.org/10.5194/wes-2021-70, in review, 2021.

Bottasso, C. L., Campagnolo, F., and Petrović, V.: Wind Tunnel Testing of Scaled Wind Turbine Models: Beyond Aerodynamics, Journal of Wind Engineering and Industrial Aerodynamics, 127, 11–28, https://doi.org/10.1016/j.jweia.2014.01.009, 2014.

Campagnolo, F., Petrović, V., Schreiber, J., Nanos, E. M., Croce, A., and Bottasso, C. L.: Wind Tunnel Testing of a Closed-Loop Wake Deflection Controller for Wind Farm Power Maximization, Journal of Physics: Conference Series, 753, https://doi.org/10.1088/1742-6596/753/3/032006, 2016.

Frehlich, R. G., and M. J. Yadlowsky.: Performance of mean-frequency estimators for Doppler radar and lidar, Journal of atmospheric and oceanic technology 11.5, 1217-1230, 1994.

Frehlich, R. G.: Effects of wind turbulence on coherent Doppler lidar performance. Journal of Atmospheric and Oceanic Technology 14.1, 54-75, 1997.

Lenschow, D. H., Volker W., and Christoph S.: Measuring second-through fourth-order moments in noisy data. Journal of Atmospheric and Oceanic technology 17.10, 1330-1347, 2000.

Sjöholm, M., Mikkelsen, T., Mann, J., Enevoldsen, K., & Courtney, M.: Spatial averaging-effects on turbulence measured by a continuous-wave coherent lidar. Meteorologische Zeitschrift, 18(3), 281-287, 2009.

Tian, W., Ozbay, A., and Hu, H.: A Wind Tunnel Study of Wind Loads on a Model Wind Turbine in Atmospheric Boundary Layer Winds, Journal of Fluids and Structures, 85, 17–26, https://doi.org/10.1016/j.jfluidstructs.2018.12.003, 2018.

---

## Author Comment (AC3)

**Author's Response**

On the second referee comment on "Modelling the Spectral Shape of Continuous-Wave Lidar Measurements in a Turbulent Wind Tunnel" by Marijn Floris van Dooren et al., Atmos. Meas. Tech. Discuss., https://doi.org/10.5194/amt-2021-233-RC2, 2021

26.11.2021

Marijn Floris van Dooren et al.

Dear Sir/Madam,

Thank you very much for taking the time to review our pre-print manuscript and for your helpful feedback and questions. We sincerely appreciate your explicit mention that this work deserves publication in Atmospheric Measurement Techniques. We will rephrase your comments in blue and include our response in black.

**General comments**

1. The repeated reference to better correlation for TI = 22% than TI = 3%... I believe the RMSE is lower for the TI = 3%, and the better $R^2$ correlation of the high TI case is just a function of the range of velocities used in the linear regression. The authors' explanation about relatively higher energy content in the low frequency range (that the lidar can resolve well) for the higher TI case is understood, but the reviewer wonders if this is really a critical piece of the puzzle or not. See my comments further below.

We checked the root-mean-square error (RMSE) for both the 3% and 22% turbulence intensity (TI) cases and confirmed they are 0.22 m/s and 0.65 m/s, respectively. Your assumption that the RMSE would be lower for the 3% TI case is therefore proven right. We agree that the differences in the $R^2$ values might also be caused by the different range of the measured values and will adjust our explanation accordingly.

2. The discussion and reasoning around Eq. (11) is a little bit sparse and not completely intuitive.

Usually it is assumed that lidar noise is a property of the measurement device itself, and thus uncorrelated to the measured flow. However, our findings show that the noise level significantly increases when going towards more energetic flows, with higher wind speeds. The dependency of $\sigma_\eta$ on $u_\infty$ and $\varepsilon$ was identified by having the best fit with respect to a manually fitted value. Please see our more elaborate reasoning as an answer to your comment at **L212**.

3. Dual-Doppler vs single-Doppler. It is not entirely clear to me that the dual-Doppler data belongs in this paper, since the main results in this article only use WS2 (because the spectral transfer function is for a single-Doppler only). I would ask that the focus on the dual Doppler reconstruction be removed or justified better. Moreover, the authors state in more than one place that the dual-Doppler reconstruction is better than either of the individual wind scanners for the u component of velocity. I don't think this is supported by Figure 17/18 or Table 4. If the dual-Doppler is NOT better, this doesn't overshadow the usefulness of a dual-Doppler technique which can resolve two components of velocity, but the text appears misleading about the $u$-component results. (I also noticed that the dual-Doppler results are not even mentioned in the abstract, which further makes me wonder: Why even have the dual Doppler results in this paper?)

It is a very fair point of the reviewer that the paper focuses on dual-Doppler too much, although the main results are based on the measurements of a single WindScanner. The research presented in the paper is based on a measurement campaign that was designed for the measurement of the two-dimensional wind speed through the wind tunnel. As this has major implications for the measurement setup, we would like to keep the notion of dual-Doppler in the paper and include it in the comparison of statistics and goodness of fit coefficients. However, the analysis of the $v$-component (**L369-384**) does not contribute enough to the main objectives of the paper and is therefore omitted.

The strong statement that the dual-Doppler reconstruction is generally performing better than the projected wind speeds based on the single WindScanners is not always correct, as you have rightfully mentioned. That is why statements in both **L338-339** and in **L427-429** have been omitted.

**Specific comments**

Section 3

**L173-174**: Does the work of Sjöholm *et al*. and Angelou *et al*. specifically describe the higher frequencies in the lidar spectra as white noise? While it may seem obvious to some, there is not strong justification given for why white noise (i.e. the Gaussian distribution) is chosen. As this is a main advance of the paper, I think a little more information is warranted.

Sjöholm *et al*. (2009) mention '…the noise induced feature at the very highest frequencies…' which implies they suggest that the highest frequencies in the spectrum are affected by noise. The type of noise (e.g. Gaussian, white noise) is not specified in this article. According to Angelou *et al*. (2012), however, the power spectral density function having an increased amplitude for the higher frequency range, is said to be '…probably due to white noise…'. Also, according to our own findings, it is a fair to assume that the noise at the higher frequency range can be classified as Gaussian, white noise, since the spectrum tends towards a horizontal line at the highest frequencies, which is a characteristic of white noise. Such a statement will be added to the paper.

**L212**: It is not self-evident that $\sigma_\eta$ should be a function of energy dissipation rate or wind speed. Is the energy dissipation and wind speed related to the decorrelation time? Is this measurement shot-noise limited? Could you explain this more?

We were not able to identify references that suggest a relationship between the lidar spectral noise level and flow parameters such as the energy dissipation rate and mean wind speed. However, we found very clear indications that such a connection does exist and would like to elaborate on it. First, we started with the assumption that the noise in a lidar measurement should be related to random fluctuations of the backscattered signal only, and that this is a property inherent to the lidar measurement principle and not to the physical properties of the turbulent flow. However, in our analysis we saw a convincing increase of the noise level for more energetic flows with higher wind speeds. We have evaluated various possible dependencies. The following lists the steps describing our empirical analysis of the lidar spectral noise estimate:

1. For each case (1a, 1b and 2a-2e) we manually tuned the lidar noise standard deviation $\sigma_\eta$ to the model in Eq. (10) for the best possible match between modelled and measured lidar spectrum.
2. With a linear regression, we then tried to identify a parameter or a combination of parameters that could best match those tuned values for $\sigma_\eta$.
3. In the end the best fit was found for the square root of the product of energy dissipation rate $\varepsilon$ and mean wind speed $\mu_u$.

Figure 1 shows the relationship of $\sigma_\eta$ with the mean wind speed $\mu_u$ and the standard deviation $\sigma_u$. The fit is not convincing, although the dimensions match.

[Figure]

**Figure 1**: *Plots of the relationship of the lidar spectral noise standard deviation $\sigma_\eta$ with the mean wind speed $\mu_u$ and the standard deviation $\sigma_u$, respectively.*

Figure 2 displays the relationship of $\sigma_\eta$ with three different definitions of the coefficient of variance $c_v$, which is like the standard deviation $\sigma_u$, but only considers the small-scale fluctuations, which are most likely to influence the lidar noise. The three plots look similar, although the absolute values are different. The fit is a significant improvement compared to the standard deviation $\sigma_u$ of the full time series. The unit matches to m/s.

[Figure]

**Figure 2**: *Plots of the relationship of the lidar spectral noise standard deviation $\sigma_\eta$ with the coefficient of variation $c_v$ calculated in different ways; Left: Difference between the modelled lidar time series (only Lorentzian filter without added noise) and the hot-wire time series. Middle: Integrated coefficient of variance of the hot-wire spectrum from $f_c$ to infinity. Right: Integrated coefficient of variance of the hot-wire spectrum from $f_{cc}$ to infinity.*

Finally, Fig. 3 shows the more convincing relationships that include the energy dissipation rate $\varepsilon$ (with the unit m$^2$/s$^3$), both alone and in combination with the mean wind speed $\mu_u$.

[Figure]

**Figure 3**: *Plots of the relationship of the lidar spectral noise standard deviation $\sigma_\eta$ with the square root of the energy dissipation rate and the product of it with the mean wind speed, respectively*.

Based on this thorough empirical analysis, we decided that there is a convincing relationship between the noise standard deviation and the energy dissipation rate in combination with the mean wind speed of the flow. Our interpretation of this phenomenon is that more energetic flows, with a higher energy dissipation rate, inhabit more pronounced fluctuations and gradients in the probe volume of the cw-lidar, which result in a higher uncertainty in the estimated wind speed that influences the small-scale fluctuations in the cw-lidar time series, regardless of the Lorentzian low-pass filter effect.

This hypothesis needs more elaboration. Most likely the white noise is partly due to shot noise of the lidar, and partly related to global flow parameters. Please note that we are not suggesting the white noise is in any way correlated to the flow in the time domain, it is mainly the notion that global flow parameters could influence the absolute value of the lidar noise standard deviation. The model is not yet complete and might rely on further flow parameters, e.g. time and length scales, and lidar parameters.

Since we are convinced that this could be an interesting finding, we would like to keep the adjusted model described by Eq. (11) in the paper. Currently the empirical analysis is not included in the paper itself, but if this is deemed necessary, we will add (part of) this elaboration as an appendix.

**Eq. (11)**: Why is gravity a relevant variable in the dimensional analysis?

We tried to make the units match by including physical constants, of which the gravitational acceleration parameter seemed to be the best candidate. However, we agree with you that it should not physically play a role in this relationship so we left it out and accept a constant with a unit instead, indicating that there might still be unidentified parameters playing a role in the estimation of the lidar spectral noise level.

Section 4.1

**L255**: I don't know if I would put so much weight on $R^2$ values here. I wonder how the RMSE compares between figures 9 and 10. The larger spread of u from a turbulent field seems to be giving higher $R^2$ even though there is clearly more absolute variation between $u_p$ and u over most of the range for the more turbulent case. If you were to run the lower TI case at a freestream velocity of both 5 m/s and 15 m/s (i.e., over the same range as shown for the high TI case), the $R^2$ of the combined data for the low TI case would be larger than for the high TI case, right?

The first referee also pointed out that the $R^2$ values of the instantaneous measurements for the 3% and 22% turbulence cases are not particularly relevant, unless the time series are smoothed with a moving average before, to filter out the larger fluctuations and improve the comparability. We have decided to follow this advice.

We used a window of 20 samples, since this way the effective averaging of the 451.7 Hz time series will yield a smoothed time series where frequencies above ~22.6 Hz are filtered out. This value is just below the lowest cut-off frequency of the lidar measurement modelled by means of the Lorentzian filter full width half maximum among the three presented cases. The effect of the smoothing on the goodness of fit coefficient for the three cases portrayed in the paper is listed in Table 1.

**Table 1**: *Improvement of the goodness of fit coefficient for the correlation between the smoothed WindScanner 2 and hot wire time series for three cases (1a, 1b and 2c).*

| Case | Figure | $R^2$ (instantaneous) | $R^2$ (smoothed) |
|------|--------|------------------------|-------------------|
| 1a | 9 | 0.656 | 0.790 |
| 1b | 10 | 0.931 | 0.957 |
| 2c | 16 | 0.950 | 0.975 |

The moving average procedure mostly benefits the goodness of fit coefficient for case 1a, while also slightly increasing the already high values of cases 1b and 2c.

In addition, we checked the root-mean-square error (RMSE) for both the 3% and 22% turbulence intensity (TI) cases and confirmed they are 0.22 m/s and 0.65 m/s, respectively.

**L284-285**: I understand that you are saying the small scales play a more dominant role for Figure 11 than Figure 12, which seems true based on the low frequency amplitudes of $S(f)$. The energy content at $f_{cc}$ is still more than 10 times lower than at lower frequencies for Fig. 11, though. Why don't you integrate Figures 11 and 12 from 0 to $f_{cc}$ and from $f_{cc}$ to infinity. See what fraction of the turbulence is not fully resolvable by the lidar and report this rather than emphasizing the difference in $R^2$ values, which doesn't seem as relevant to me.

We followed up on your idea and integrated the power spectral density of the time series of WindScanner 2 to yield an equivalent variance of both the low and high frequency ranges, as a measure of how much energy is contained within the respective scales. We use the fact that the integral of the power spectral density function over frequency yields the variance. We define $\sigma_{u_l}$ and $\sigma_{u_h}$ for the standard deviation of the low and high frequency range of the time series $u_p$, respectively:

$$\sigma_{u_l}^2 = \int_0^{f_{cc}} fS(f)df$$

$$\sigma_{u_h}^2 = \int_{f_{cc}}^{\infty} fS(f)df$$

The resulting quantities, as well as their ratio, can be read off Table 2:

**Table 2**: *Comparison of the equivalent variance of the low and high frequency ranges of the WindScanner 2 projected wind speed time series.*

| Case | TI [%] | $\sigma_{u_l}^2$ [m²/s²] | $\sigma_{u_h}^2$ [m²/s²] | $\frac{\sigma_{u_h}^2}{\sigma_{u_l}^2+\sigma_{u_h}^2}$ [%] |
|------|--------|---------------------------|---------------------------|------------------------------------------------------------|
| 1a | 3 | 0.074 | 0.019 | 20.4 |
| 1b | 22 | 4.9 | 0.19 | 3.8 |

As suspected, the contribution to the variance by the scales that are not fully resolvable by the lidar is higher for the case of 3% turbulence, with a ratio of 20.4% to 3.8%. These numbers will be reported in the paper. However, as explained in the response at **L255**, we will not completely omit the correlation plots and the reported $R^2$ values.

**L289**: I was expecting this line to say, "A possible reason is the insufficiency of the Full Width at Half Maximum metric to characterize the effective probe length." Don't you agree? What about the implicit assumption that the turbulence is isotropic, could this also be a possible culprit?

Thank you for providing an alternative explanation for the much lower frequencies at which the lidar spectrum deviates from the hot wire spectrum. We agree that a likely reason is indeed the insufficiency of the full width half maximum definition, however we would still like to also mention the misalignment of the probe volume with the $x$-axis, which could invalidate the assumption of isotropic turbulence along the line-of-sight. We reformulated the sentence as follows: 'Possible reasons for this are the insufficiency of the Full Width at Half Maximum metric to characterise the effective probe length, and the invalidity of the assumption of isotropic turbulence, combined with the misalignment between the line-of-sight and the $x$-direction'.

Section 4.2

**L325**: This is the first time you've mentioned 1 Hz averaged time series. Could you please give a brief mention of why you perform this time averaging (I assume to get out of the small eddy range that can't be resolved by the lidar).

You are right about the reason for the 1 Hz averaged time series correlation. The goal is to eliminate the range of small-scale turbulence that cannot be resolved by the cw-lidar. This mention is now included in the paper.

**L331**: Can you comment on why the green line is not the highest for the 1 Hz data. Is it that at 452 Hz, the two lidars are both filtering small-scale turbulence and thus agree quite closely compared to the unfiltered hot-wire, but at 1 Hz, both the hot-wire and lidar are on more even playing field and can both resolve all the scales?

After applying a moving average window, as explained in the response at **L255**, the green line (comparing the two WindScanners with each other) is neither the best for the 451.7 Hz time series nor for the 1 Hz time series. Although the WindScanners are theoretically identical devices, there are tolerances in the optical system that may cause differences in the measurement. This will be mentioned in the corresponding section of the paper. Your explanation about the hot wire and lidar being on a more even playing field at lower sampling rates could also apply, though.

**L338**: The lowest errors appear to be found for the blue line not the black line, and a quick subtraction of the columns in Table 4 suggests that the mean difference between WS1 and HW is smaller than between WS and HW. Please revise this statement or justify it. Is the mean error of the dual-Doppler reconstruction related to the fact that the hot-wire only measures one component? Why is validating the dual-Doppler reconstruction given so much weight in this paper?

After reconsidering the plots in Fig. 18 of the paper, we agree with your observation. However, the order of the lines changed slightly after performing the moving average procedure as explained before. On top of that we now replaced the relative mean wind speed difference with both mean average error (MAE) and root-mean-square error (RMSE) plots, which are more common ways to address errors. The paragraph describing the figures is updated accordingly.

It is likely that the mean error of the dual-Doppler reconstructed wind speed with respect to the hot wire is related to the one-dimensional nature of the hot wire measurement. However, considering the usually very low lateral wind speed components in the wind tunnel, the magnitude of the errors found cannot be explained by that exclusively. We believe that a larger contribution is coming from the heterogeneity of the wind conditions over the effective probe volume of the lidars, compared to the single point at which the hot wire is mounted, and over the 7 cm separation between the lidars' focus point and the hot wire location.

The dual-Doppler reconstruction should play a lesser role in the paper, as we have described in the answer to your 3rd general comment. We would like to have this result in the plot as a reference but will put less emphasis on it.

**L342**: You say there is a "bias between WindScanner 1 and 2, which increases linearly with the mean wind speed". This is not obvious from the plot except moving from $\mu_u \sim 2$ m/s to $\mu_u \sim 5$ m/s where the gap widens between red and blue. Please revise or justify.

The statement of the 'linear increase of the error' can be justified when multiplying the relatively constant percentual increase (over the range between 5 m/s and 11 m/s) with the absolute wind speed values. However, we acknowledge that this is a confusing statement that has been rephrased as a 'relative bias'.

**L350**: You say, "they do not have physical significance". Please clarify your statement about physical significance as this is clearly a physical phenomenon in the flow that is being resolved by both measurement systems.

We agree that it is incorrect to label the frequency peaks as 'no physical significance'. The first referee also pointed this out. We changed the wording to 'As they result from external gust variations, these peaks are not deemed to be due to turbulence and have a much larger scale than the lowest scales detectable by the WindScanners'.

**Table 5**: I wonder if the ratios of $f_c/f_{cc}$ in this table are possibly more important in the long run than the 0.5 coherence observation, since in a real application of this technique, you will not have a reference instrument to calculate coherence, right? Would it be appropriate to suggest that the effective probe volume given by the FWHM could be at most an order of magnitude in error based on this data?

We agree with the conclusion that the Full Width at Half Maximum is not sufficient as a length scale for defining the extent of the spatial filtering effect in the probe volume. The noticeable filtering effect indeed occurs over a range that is around an order of magnitude larger than the 'classic' probe length definition. However, we think it would be a too strong statement based on the limited data set to define an 'effective probe length' with an order of magnitude larger than the other definition. We did add a mention in the Conclusion section of the paper about it.

**L375**: I think this is a good conclusion to draw. It looks like the amplitude of the protocol-induced gust is larger in the u rather than $v$-direction – could this be a reason why the triple repetition is being lost in Figure 22? Reading further, I see that these differences are quantified in Table 6. If you believe my argument, I think you could comment on how the fact that $\sigma_v/\sigma_u \ll 100\%$ might be related to your conclusion in line 375.

It is indeed true that the amplitude of the induced gust is much larger in the $u$-component, which is the variable meant to be influenced by the active grid protocol. Although designing active grid protocols with the purpose of simulating the $v$-component should be possible, it was not applied in this case. However, we disagree that the triple gust is completely lost in Fig. 22, since there is still a visible signature, albeit much less pronounced.

As we have stated in the answer to your 3rd general comment, we decided to omit the analysis of the $v$-component, since the dual-Doppler reconstruction is not the main objective of this paper.

Section 4.3

**L392**: You say, "the latter curve is not valid for large-scale structures". Just to clarify, is this because it is only derived for the inertial subrange?

Your presumption is correct; the analytical formulas for the modelled lidar spectrum are specifically derived for the inertial sub-range. This notion is now included.

**L415**: The potential application is very interesting and seems worthwhile. You have used the word "atmospheric" twice in the last three paragraphs. From the introduction of the paper, I was under the impression that you want to use the dual lidar technology in wind tunnel studies of wind turbine configurations? Could you clarify here (and in the abstract/introduction) if your aim is for wind tunnel or field measurements (or both)?

Actually, we do not exclusively reserve the word 'atmospheric' for wind conditions in the free field, but also for the flow through our wind tunnel. We would like to refer to **L20** where we state that 'Existing wind tunnels can simulate the atmospheric boundary layer through passive flow manipulation…'. However, we understand that the definition 'atmospheric' might be confusing for the general statement made here, so we chose to omit it from this sentence.

Having said this, we believe that it would be an interesting comparison whether the models presented in our paper would also work for atmospheric flow measured in the free field. We have added the following sentence at the end of the Abstract of the paper: 'Although the models were developed on the basis of wind tunnel measurements, the application on free field measurements should be possible as well.'

Section 5

**L427**: In reference to 1.1%, you say that the dual-Doppler gives "lower spread". However, the 1.1% comes from an analysis of mean error, not scatter. Could you clarify this wording?

Thank you for correcting the wording here, where we should have used 'mean error' instead of 'spread'. However, since we have now also included the analysis of the RMSE of the time series, there will be additional lines addressing that as well.

**Technical comments**

**Fig. 1**: It might be more useful if it included a zoom in of the nozzle with the active grid.

We have considered providing a close-up of the active grid. However, for the interpretation of the results in this paper, we believe that it is more valuable to know what the inside of the wind tunnel looked like during the measurement. We would prefer to refer to Kröger *et al*. (2018), who described the wind tunnel and the active grid in more detail, and included several photos.

**L68**: Add "to" before reproduce.

We added the preposition 'to' to this sentence.

**Table 2**: I think it would be appropriate to give the names of the variables in Table 2 and not just the symbols.

Since this table contains variables that are not introduced until later, the names of the variables have been added.

**L140**: You mention that $L$ is the probe length twice.

The probe length $L$ was introduced twice. We removed the redundancy and merged the two respective sentences into one.

**Fig. 11/12**: Could you note in the caption that $f_{cc}$ will be defined later in Section X?

We added the notion that the variable $f_{cc}$ will be defined later in Subsect. 4.2 at the end of **L265**, as opposed to in the captions of both Fig. 11 and Fig. 12, to avoid redundancy.

**L327-328**: No need to describe what the different colored lines mean since it's in the figure.

We agree to remove the redundant lines that describe the meaning of the graph colours, as the figure caption should be sufficient.

**L346**: No need to describe the line colors in the text.

We agree to the removal of the graph colour definitions here, since at this point in the paper it should have already been clear from Fig. 11 and Fig. 12.

**L427**: You write "down to −1.1%." Can you say "within 1.1%" instead to be more precise?

We agree with you that changing the wording 'down to −1.1%' to 'within 1.1%' makes sense. However, since we are now using the MAE and the RMSE instead of the straight difference, the paragraph describing these results has changed.

**References**

Angelou, N.; Mann, J.; Sjöholm, M. & Courtney, M.: Direct Measurement of the Spectral Transfer Function of a Laser based Anemometer, Review of Scientific Instruments, 83, https://doi.org/10.1063/1.3697728, 2012.

Kröger, L., Frederik, J., van Wingerden, J. W., Peinke, J., and Hölling, M.: Generation of User Defined Turbulent Inflow Conditions by an Active Grid for Validation Experiments, Journal of Physics: Conference Series, 1037, https://doi.org/10.1088/1742-6596/1037/5/052002, 2018.

Sjöholm, M.; Mikkelsen, T.; Mann, J.; Enevoldsen, K. & Courtney, M.: Spatial Averaging Effects of Turbulence Measured by a Continuous-Wave Coherent Lidar, Meteorologische Zeitschrift, 18, https://doi.org/10.1127/0941-2948/2009/0379, 2009.

---

## Referee Report (RR1)

The authors have significantly improved the manuscript and thoroughly addressed the review comments.

The only point of question is the development of Equation 11, which has been presented thoroughly and explicitly in the author response but has not been discussed in the manuscript. The reviewer suggests that Figure 3 (of amt-2021-233-author\_response-version1.pdf) could be included in the paper. Further, a statement that there was no correlation between  $\sigma_{\eta}$  and the ambient temperature, which could hypothetically be a function of how hard the wind tunnel blower has to work to produce given global flow conditions, would remove the remote possibility that internal lidar noise is somehow a factor.

The reviewer also wonders if the results of the empirical analysis, which indicate that the white noise is due to shot noise AND global flow parameters, do already have some basis in literature. The global flow parameters will influence the de-correlation time of the lidar return. Specifically, turbulence level and scanning speed are known to influence de-correlation time [Lindelöw, 2008, Appendix B; others], and these two parameters could have relation to the energy dissipation rate and mean flow velocity, respectively, that were identified by the present authors as influencers of the noise magnitude (the mean flow velocity might be considered a surrogate for scan speed in the case of the static, off-axis scan configuration considered by the authors). In cw lidar, the de-correlation time affects the width of the Doppler spectra, which may affect the precision of the parameter estimation process used to determine the line-of-sight velocity.

**References:**

Lindelöw, P. Fiber Based Coherent Lidars for Remote Wind Sensing Ph. D. Diss. thesis Danish Technical University, 2008.

---

## Author Response (AR2)

**Author's Response**

On the second referee comment during the second review round of the manuscript "Modelling the Spectral Shape of Continuous-Wave Lidar Measurements in a Turbulent Wind Tunnel" by Marijn Floris van Dooren et al., Atmos. Meas. Tech. Discuss., https://amt.copernicus.org/preprints/amt-2021-233/#discussion, 2021.

03.02.2022

Marijn Floris van Dooren et al.

Dear Sir/Madam,

Thank you very much for reviewing our revised manuscript again and for your valuable feedback. We isolated and rephrased your comments in blue and included our response in black.

**Comments**

**1**: The authors have significantly improved the manuscript and thoroughly addressed the review comments. The only point of question is the development of Eq. (11), which has been presented thoroughly and explicitly in the author's response but has not been discussed in the manuscript. The reviewer suggests that Fig. 3 (of amt-2021-233-author\_response-version1.pdf) could be included in the paper.

We acknowledge that the justification of Eq. (11) might still not be sufficiently covered in the paper. Therefore, we now added an Appendix B to the manuscript, which describes the development of said equation in more detail, including a plot with the correlation between noise standard deviation and the term including energy dissipation rate and mean wind speed.

**2**: Further, a statement that there was no correlation between  $\sigma_{\eta}$  and the ambient temperature, which could hypothetically be a function of how hard the wind tunnel blower has to work to produce given global flow conditions, would remove the remote possibility that internal lidar noise is somehow a factor.

We have indeed recorded the ambient temperature in the wind tunnel before every 10-minute measurement, which was in the range between 17.8°C and 19.1°C for the measurements presented in the manuscript. No significant correlation between the ambient temperature and the value of  $\sigma_{\eta}$  could be identified, of which Fig. 1 is proof.

**Figure 1**: Plot of the relationship of the lidar spectral noise standard deviation  $\sigma_{\eta}$  with the ambient temperature *T* inside the wind tunnel.

Following the reviewer's suggestion there is a statement about this added to the manuscript (L229-L230), however, the plot in Fig. 1 is not deemed important enough to include in the paper.

**3**: The reviewer also wonders if the results of the empirical analysis, which indicate that the white noise is due to shot noise and global flow parameters, do already have some basis in literature. The global flow parameters will influence the de-correlation time of the lidar return. Specifically, turbulence level and scanning speed are known to influence de-correlation time (Lindelöw, 2008, Appendix B; others), and these two parameters could have relation to the energy dissipation rate and mean flow velocity, respectively, that were identified by the present authors as influencers of the noise magnitude (the mean flow velocity might be considered a surrogate for scan speed in the case of the static, off-axis scan configuration considered by the authors). In cw-lidar, the de-correlation time affects the width of the Doppler spectra, which may affect the precision of the parameter estimation process used to determine the line-of-sight velocity.

Lindelöw (2008) discusses the effects of changing aerosol backscatter correlation time on measured spectra obtained with a cw-lidar. In Appendix B, it is mentioned that the width of a single measured Doppler spectrum is inversely proportional to the correlation duration  $\tau$  unless the spectral width is dominated by spread due to different speeds in the ensemble of many spectra sampled in the sampling volume, that is, by turbulence. It is stated that the global flow parameters such as mean flow velocity and dissipation rate will in general influence the de-correlation time of the lidar return. Specifically, turbulence level and scanning speed are known factors to influence de-correlation time.

For our study in the wind tunnel, the WindScanner cw-lidars, based on ZephIR technology, are configured to sample 200,000 spectra per second, based on a sampling time per spectrum of 5  $\mu$ s. The lidar's signal-to-noise ratio after sampling ~443 spectra per measurement (at sampling rate of 451.7 Hz) is found to be sufficiently high that the observed spectral spread can be assumed to be dominated by turbulence in the flow, and not due to changing de-correlation times due to flow parameters over individual single 5  $\mu$ s sampling time. Presumably, the biggest effect of a short de-correlation in the backscatter signal would result in lower signal-to-noise ratios, but to a lesser degree in measured spectral width observations. Please also note that we have been intermittently applying generous seeding in the wind tunnel (See L98-L103 of the manuscript), to guarantee a sufficient amount of homogeneously distributed particles in our measurement region.

Even if a specific flow condition would influence the de-correlation time, it would probably not affect the measured Doppler shift estimation to a large extent, since it is determined from the mean Doppler shift from 443 spectra in case of 451.7 Hz sampling. The width of the ensemble-averaged spectral as used per measurement will be dominated by turbulence in the probe volume, as opposed to the width of the individual spectra. Where the individually sampled spectra and their width may be affected by changes in de-correlation time due to specific flow conditions, the ensemble-averaged spectra, from which we determine the effective Doppler shift by fitting a centroid mean value, will probably not.

**References**

Lindelöw, P. Fiber Based Coherent Lidars for Remote Wind Sensing. Ph.D. Dissertation Thesis. Danish Technical University, 2008.

**Changes in the revised manuscript**

Here the changes that have been implemented in the manuscript are listed point-by-point.

**03.02.2021**

Marijn Floris van Dooren et al.

**Methodology, Part II: The Physical Models**

- A statement has been added addressing the investigation of the dependency of the standard deviation of the noise on the standard deviation of the mean wind speed and the ambient temperature (L228-L230).
- A reference to the newly added Appendix B has been added (L236-L237).

**Appendix A**

• An introductory sentence has been added to Appendix A.

**Appendix **B**

• The newly added Appendix B contains a more elaborate explanation of the development of the expression in Eq. (11), including a plot.

**References**

• The references to the recently published papers by Berger et al. (2021) and Neuhaus et al. (2021) have been updated accordingly.

Berger, F., Onnen, D., Schepers, J. G., and Kühn, M.: Experimental Analysis of Radially Resolved Dynamic Inflow Effects due to Pitch Steps, Wind Energ. Sci., 6, 1341–1361, https://doi.org/10.5194/wes-6-1341-2021, 2021.

Neuhaus, L., Berger, F., Peinke, J., and Hölling, M.: Exploring the Capabilities of Active Grids, Experiments in Fluids, 62, https://doi.org/10.1007/s00348-021-03224-5, 2021.